# About the Cost of Central Privacy in Density Estimation

**Clément Lalanne**                                                   *clement.lalanne@ens-lyon.fr*
*Univ. Lyon, ENS Lyon, UCBL, CNRS, Inria, LIP, F-69342, Lyon Cedex 07, France*

**Aurélien Garivier**                                                 *aurelien.garivier@ens-lyon.fr*
*Univ. Lyon, ENS Lyon, UMPA UMR 5669, 46 allée d'Italie, F-69364, Lyon cedex 07*

**Rémi Gribonval**                                                    *remi.gribonval@inria.fr*
*Univ. Lyon, ENS Lyon, UCBL, CNRS, Inria, LIP, F-69342, Lyon Cedex 07, France*

**Reviewed on OpenReview:** `https://openreview.net/forum?id=uq29MIWvIV`

## Abstract

We study non-parametric density estimation for densities in Lipschitz and Sobolev spaces, and under *central* privacy. In particular, we investigate regimes where the privacy budget is *not* supposed to be constant. We consider the classical definition of central differential privacy, but also the more recent notion of central concentrated differential privacy. We recover the result of Barber & Duchi (2014) stating that histogram estimators are optimal against Lipschitz distributions for the $L^2$ risk and, under regular differential privacy, we extend it to other norms and notions of privacy. Then, we investigate higher degrees of smoothness, drawing two conclusions: First, and contrary to what happens with *constant* privacy budget (Wasserman & Zhou, 2010), there *are* regimes where imposing privacy degrades the regular minimax risk of estimation on Sobolev densities. Second, so-called projection estimators are near-optimal against the same classes of densities in this new setup with pure differential privacy, but contrary to the constant privacy budget case, it comes at the cost of relaxation. With zero concentrated differential privacy, there is no need for relaxation, and we prove that the estimation is optimal.

## 1 Introduction

The communication of information built on users' data leads to new challenges, and notably privacy concerns. It is now well documented that the release of various quantities can, without further caution, have disastrous repercussions (Narayanan & Shmatikov, 2006; Backstrom et al., 2007; Fredrikson et al., 2015; Dinur & Nissim, 2003; Homer et al., 2008; Loukides et al., 2010; Narayanan & Shmatikov, 2008; Sweeney, 2000; Gonon et al., 2023; Wagner & Eckhoff, 2018; Sweeney, 2002). In order to address this issue, differential privacy (DP) (Dwork et al., 2006b) has become the gold standard in privacy protection. The idea is to add a proper layer of randomness in order to hide each user's data. It is notably used by the US Census Bureau (Abowd, 2018), Google (Erlingsson et al., 2014), Apple (Thakurta et al., 2017) and Microsoft (Ding et al., 2017), among many others.

As with other forms of communication or processing constraints (Barnes et al., 2019; 2020; Acharya et al., 2021a;b;c;d), privacy recently gained a lot of attention from the statistical and theoretical machine learning communities. At this point, the list of interesting publications is far too vast to be exhaustive, but here is a sample : Wasserman & Zhou (2010) is the first article to consider problems analogous to the ones presented in this article. It notably studies the problem of nonparametric density estimation, to which we provide many complements. Duchi et al. (2014; 2013; 2016); Barber & Duchi (2014); Acharya et al. (2021e); Lalanne et al. (2023a) present general frameworks for deriving minimax lower-bounds under privacy constraints. Many parametric problems have already been studied, notably in Acharya et al. (2018; 2021e); Karwa & Vadhan (2018); Kamath et al. (2019); Biswas et al. (2020); Lalanne et al. (2022; 2023b); Kamath et al. (2022); Singhal (2023). Recently, some important contributions were made. For instance, Asi et al. (2023) sharply characterized the equivalence between private estimation and robust estimation with the inverse sensitivity

mechanism (Asi & Duchi, 2020a;b), and Kamath et al. (2023) detailed the bias-variance-privacy trilemma, proving in particular the necessity of adding bias, even on distributions with bounded support, for many private estimation problems.

We address here the problem of privately estimating a probability density, which fits in this line of work. Given $\mathbf{X} := (X_1, \ldots, X_n) \sim \mathbb{P}_\pi^{\otimes n}$, where $\mathbb{P}_\pi$ refers to a distribution of probability that has a density $\pi$ with respect to the Lebesgue measure on $[0, 1]$, how to estimate $\pi$ privately? Technically, what metrics or hypothesis should be set on $\pi$? What is the cost of privacy? Are the methods known so far optimal? Such are the questions that are investigated in the rest of this article.

## 1.1 Related work

Non-parametric density estimation has been an important topic of research in statistics for many decades now. Among the vast literature on the topic, let us just mention the important references Györfi et al. (2002); Tsybakov (2009).

Recently, the interest for private statistics has shone a new light on this problem. Remarkable early contributions (Wasserman & Zhou, 2010; Hall et al., 2013) adapted histogram estimators, so-called projection estimators and kernel estimators to satisfy the privacy constraint. They conclude that the minimax rate of convergence, $n^{-2\beta/(2\beta+1)}$, where $n$ is the sample size and $\beta$ is the (Sobolev) smoothness of the density, is not affected by *central* privacy (also known as *global* privacy). However, an important implicit hypothesis in this line of work is that $\epsilon$, the parameter that decides how private the estimation needs to be, is supposed not to depend on the sample size. This hypothesis may seem disputable, and more importantly, it fails to precisely characterize the tradeoff between utility and privacy.

Indeed, differential privacy gives guarantees on *how hard* it is to tell if a specific user was part of the dataset. Despite the fact that one could hope to leverage the high number of users in a dataset in order to increase the privacy w.r.t. each user, previous studies (Wasserman & Zhou, 2010; Hall et al., 2013) cover an asymptotic scenario with respect to the number of samples $n$, for *fixed $\epsilon$*. In contrast, our study highlights new behaviors for this problem. For each sample size, we emphasize the presence of two regimes: when the order of $\epsilon$ is larger than some threshold (dependent on $n$) that we provide, privacy can be obtained with virtually no cost; when $\epsilon$ is smaller than this threshold, it is the limiting factor for the accuracy of estimation.

To the best of our knowledge, the only piece of work that studies this problem under *central* privacy when $\epsilon$ is not supposed constant is Barber & Duchi (2014). They study histogram estimators on Lipschitz distributions for the integrated risk. They conclude that the minimax risk of estimation is $\max\left(n^{-2/3} + (n\epsilon)^{-1}\right)$, showing how small $\epsilon$ can be before the minimax risk of estimation is degraded. Our article extends such results to high degrees of smoothness, to other definitions of *central* differential privacy, and to other risks.

In the literature, there exist other notions of privacy, such as the much stricter notion of *local* differential privacy. Under this different notion of privacy, the problem of non-parametric density estimation has already been extensively studied. We here give a few useful bibliographic pointers. A remarkable early piece of work Duchi et al. (2016) has brought a nice toolbox for deriving minimax lower bounds under local privacy that has proven to give sharp results for many problems. As a result, the problem of non-parametric density estimation (or its analogous problem of non-parametric regression) has been extensively studied under local privacy. For instance, Butucea et al. (2019) investigates the elbow effect and questions of adaptivity over Besov ellipsoids. Kroll (2021) and Schluttenhofer & Johannes (2022) study the density estimation problem at a given point with an emphasis on adaptivity. Universal consistency properties have recently been derived in Györfi & Kroll (2023). Analogous regression problems have been studied in Berrett et al. (2021) and in Györfi & Kroll (2022). Finally, the problem of optimal non-parametric testing has been studied in Lam-Weil et al. (2022).

## 1.2 Contributions

In this article, we investigate the impact of *central* privacy when the privacy budget is not constant. We treat multiple definitions of *central* privacy and different levels of smoothness for the densities of interest.

In terms of upper-bounds, we analyze histogram and projection estimators at a resolution that captures the impact of the privacy and smoothness parameters. We also prove new lower bounds using the classical

packing method combined with new tools that characterize the testing difficulty under central privacy from Acharya et al. (2021e); Kamath et al. (2022); Lalanne et al. (2023a).

In particular, for Lipschitz densities and under pure differential privacy, we recover the results of Barber & Duchi (2014) with a few complements. We then extend the estimation on this class of distributions to the context of concentrated differential privacy (Bun & Steinke, 2016), a more modern definition of privacy that is compatible with stochastic processes relying on Gaussian noise. We finally investigate higher degrees of smoothness by looking at periodic Sobolev distributions. The main results are summarized in Table 1.2.

| | $\epsilon$-**DP** Equation (1) | $\rho$-**zCDP** Equation (2) |
|---|---|---|
| **Lipschitz** Equation (4) | Upper-bound: $O\left(\max\left\{n^{-2/3}, (n\epsilon)^{-1}\right\}\right)$ (Barber & Duchi, 2014) & Theorem 1 <hr> Lower-bounds: -Pointwise: $\Omega\left(\max\left\{n^{-2/3}, (n\epsilon)^{-1}\right\}\right)$ Theorem 2 & Corollary 1 -Integrated: $\Omega\left(\max\left\{n^{-2/3}, (n\epsilon)^{-1}\right\}\right)$ (Barber & Duchi, 2014) & Theorem 3 | Upper-bound: $O\left(\max\left\{n^{-2/3}, (n\sqrt{\rho})^{-1}\right\}\right)$ Theorem 1 <hr> Lower-bounds: -Pointwise: $\Omega\left(\max\left\{n^{-2/3}, (n\sqrt{\rho})^{-1}\right\}\right)$ Theorem 2 & Corollary 1 -Integrated: $\Omega\left(\max\left\{n^{-2/3}, (n\sqrt{\rho})^{-1}\right\}\right)$ Theorem 3 |
| **Periodic Sobolev** Smoothness $\beta$ Equation (9) | Upper-bounds: -Pure DP: $O\left(\max\left\{n^{-\frac{2\beta}{2\beta+1}}, (n\epsilon)^{-\frac{2\beta}{\beta+3/2}}\right\}\right)$ Theorem 4 -Relaxed: $\max\left\{n^{-\frac{2\beta}{2\beta+1}}, \left(\frac{n\epsilon}{\sqrt{\ln(1.25/\delta)}}\right)^{-\frac{2\beta}{\beta+1}}\right\}$ Section 4.4 <hr> Lower-bound: $\Omega\left(\max\left\{n^{-\frac{2\beta}{2\beta+1}}, (n\epsilon)^{-\frac{2\beta}{\beta+1}}\right\}\right)$ Theorem 5 | Upper-bound: $O\left(\max\left\{n^{-\frac{2\beta}{2\beta+1}}, (n\sqrt{\rho})^{-\frac{2\beta}{\beta+1}}\right\}\right)$ Theorem 4 <hr> Lower-bound: $\Omega\left(\max\left\{n^{-\frac{2\beta}{2\beta+1}}, (n\sqrt{\rho})^{-\frac{2\beta}{\beta+1}}\right\}\right)$ Theorem 5 |

Table 1: Summary of the results

The paper is organized as follows. The required notions regarding *central* differential privacy are recalled in Section 2. Histogram estimators and projection estimators are respectively studied in Section 3, on Lipschitz densities, and in Section 4, on periodic Sobolev densities. A short conclusion in provided in Section 5.

## 2 Central Differential Privacy

We recall in this section some useful notions of *central* privacy. Here, $\mathcal{X}$ and $n$ refer respectively to the sample space and to the sample size.

Given two datasets $\mathbf{X} = (X_1, \ldots, X_n), \mathbf{Y} = (Y_1, \ldots, Y_n) \in \mathcal{X}^n$, the *Hamming* distance between $\mathbf{X}$ and $\mathbf{Y}$ is defined as

$$d_{\mathrm{ham}}(\mathbf{X}, \mathbf{Y}) = \sum_{i=1}^{n} \mathbb{1}_{X_i \neq Y_i} .$$

Given $\epsilon > 0$ and $\delta \in [0, 1)$, a randomized mechanism $\mathfrak{M} : \mathcal{X}^n \to \mathrm{codom}(\mathfrak{M})$ (for *codomain* or image of $\mathfrak{M}$) is $(\epsilon, \delta)$-differentially private (or $(\epsilon, \delta)$-DP) (Dwork et al., 2006b;a) if for all $\mathbf{X}, \mathbf{Y} \in \mathcal{X}^n$ and all measurable $S \subseteq \mathrm{codom}(\mathfrak{M})$:

$$d_{\mathrm{ham}}(\mathbf{X}, \mathbf{Y}) \leq 1 \implies \mathbb{P}_{\mathfrak{M}}(\mathfrak{M}(\mathbf{X}) \in S) \leq e^{\epsilon} \mathbb{P}_{\mathfrak{M}}(\mathfrak{M}(\mathbf{Y}) \in S) + \delta , \tag{1}$$

where $d_{\mathrm{ham}}(\cdot, \cdot)$ denotes the Hamming distance on $\mathcal{X}^n$.

In order to sharply count the privacy of a composition of many Gaussian mechanisms (see Abadi et al. (2016)), privacy is also often characterized in terms of Renyi divergence (Mironov, 2017). Nowadays, it seems that all these notions tend to converge towards the definition of *zero concentrated differential privacy* (Dwork &

Rothblum, 2016; Bun & Steinke, 2016). Given $\rho \in (0, +\infty)$, a randomized mechanism $\mathfrak{M} : \mathcal{X}^n \to \mathrm{codom}\,(\mathfrak{M})$ is $\rho$-zero concentrated differentially private ($\rho$-zCDP) if for all $\mathbf{X}, \mathbf{Y} \in \mathcal{X}^n$,

$$d_{\mathrm{ham}}\,(\mathbf{X}, \mathbf{Y}) \le 1 \implies \forall 1 < \alpha < +\infty, \mathrm{D}_\alpha\,(\mathfrak{M}(\mathbf{X}) \| \mathfrak{M}(\mathbf{Y})) \le \rho\alpha \tag{2}$$

where $\mathrm{D}_\alpha\,(\cdot \| \cdot)$ denotes the Renyi divergence of level $\alpha$, defined when $\alpha > 1$ as

$$\mathrm{D}_\alpha\,(\mathbb{P} \| \mathbb{Q}) := \frac{1}{\alpha - 1} \ln \int \left(\frac{d\mathbb{P}}{d\mathbb{Q}}\right)^{\alpha - 1} d\mathbb{Q}\,.$$

For more details, we recommend referring to the excellent article van Erven & Harremoës (2014).

There are links between $(\epsilon, \delta)$-DP and $\rho$-zCDP. For instance, if a mechanism is $\rho$-zCDP, then (Bun & Steinke, 2016, Proposition 3) it is $(\epsilon, \delta)$-DP for a collection of $(\epsilon, \delta)$'s that depends on $\rho$. Conversely, if a mechanism is $(\epsilon, 0)$-DP, then (Bun & Steinke, 2016, Proposition 4) it is also $\epsilon^2/2$-zCDP.

Given a deterministic function $f$ mapping a dataset to a quantity in $\mathbb{R}^d$, the Laplace mechanism (Dwork et al., 2006b) and Gaussian mechanism (Bun & Steinke, 2016) are two famous ways to turn $f$ into a private mechanism. Defining the $l_1$ sensitivity of $f$ as

$$\Delta_1 f := \sup_{\mathbf{X}, \mathbf{Y} \in \mathcal{X}^n : d_{\mathrm{ham}}(\mathbf{X}, \mathbf{Y}) \le 1} \left\| f(\mathbf{X}) - f(\mathbf{Y}) \right\|_1\,,$$

the Laplace mechanism instantiated with $f$ and $\epsilon > 0$ is defined as

$$\mathbf{X} \mapsto f(\mathbf{X}) + \frac{\Delta_1 f}{\epsilon} \mathcal{L}(I_d)\,,$$

where $\mathcal{L}(I_d)$ refers to a random vector of dimension $d$ with independent components that follow a centered Laplace distribution of parameter 1. Notice that we took the liberty to use the same notation for the random variable and for its distribution. We made this choice for brevity, and because it does not really create any ambiguity. It is $(\epsilon, 0)$-DP (simply noted $\epsilon$-DP) (Dwork et al., 2006a;b). Likewise, defining the $L^2$ sensitivity of $f$ as

$$\Delta_2 f := \sup_{\mathbf{X}, \mathbf{Y} \in \mathcal{X}^n : d_{\mathrm{ham}}(\mathbf{X}, \mathbf{Y}) \le 1} \left\| f(\mathbf{X}) - f(\mathbf{Y}) \right\|_2\,,$$

the Gaussian mechanism instantiated with $f$ and $\rho > 0$ is defined as

$$\mathbf{X} \mapsto f(\mathbf{X}) + \frac{\Delta_2 f}{\sqrt{2\rho}} \mathcal{N}(0, I_d)\,,$$

where $\mathcal{N}(0, I_d)$ refers to a random vector of dimension $d$ with independent components that follow a centered Normal distribution of variance 1. It is $\rho$-zCDP (Bun & Steinke, 2016).

**A quick word on local privacy.** Central privacy comes with the hypothesis of a trusted aggregator (also known as a curator, hence the alternative name a "trusted curator model" for central privacy, which is also known under the name *global privacy*) that sees the entire dataset, and builds an estimator with it. Only the produced estimator is private. In order to give an example, this is like having a datacenter that stores all the information about the users of a service, but only outputs privatized statistics.

Local privacy on the other hand does not make that hypothesis. Each piece of data is anonymized locally (on the user's device) and then it is communicated to an aggregator. Any locally private mechanism is also centrally private, but the converse is not true.

At first, local privacy can seem more appealing : it is indeed a stronger notion of privacy. However, it degrades the utility *much more* than central privacy. As a result, both notions are interesting, and the use of one or the other must be weighted for a given problem. This work focuses on the *central* variant.

## 3   Histogram Estimators and Lipschitz Densities

Histogram estimators approximate densities with a piecewise continuous function by counting the number of points that fall into each bin of a partition of the support. Since those numbers follow binomial distributions, the study of histogram estimators is rather simple. Besides, they are particularly interesting when privacy is required, since the sensitivity of a histogram query is bounded independently of the number of bins. They were first studied in this setup in Wasserman & Zhou (2010), while Barber & Duchi (2014) provided new lower-bounds that did not require a constant privacy budget.

As a warm-up, this section proposes a new derivation of known results in more modern lower-bounding frameworks (Acharya et al., 2021e; Kamath et al., 2022; Lalanne et al., 2023a), and then extends these upper-bounds and lower-bounds to the case of zCDP. Furthermore, it also covers the pointwise risk as well as the infinite-norm risk.

Let $h > 0$ be a given bandwidth or binsize. In order to simplify the notation, we suppose without loss of generality that $1/h \in \mathbb{N} \setminus \{0\}$ (if the converse is true, simply take $h' = 1/\lceil 1/h \rceil$ where $\lceil x \rceil$ refers to the smallest integer bigger than $x$). $[0,1]$ is partitioned in $\frac{1}{h}$ sub-intervals of length $h$, which are called the bins of the histogram. Let $Z_1, \dots, Z_{1/h}$ be independent and identically distributed random variables with the same distribution as a random variable $Z$ that is supposed to be centered and to have a finite variance. Given a dataset $\mathbf{X} = (X_1, \dots, X_n)$, the (randomized) histogram estimator is defined for $x \in [0,1]$ as

$$\hat{\pi}^{\text{hist}}(\mathbf{X})(x) := \sum_{b \in \text{bins}} \mathbb{1}_b(x) \frac{1}{nh} \left( \sum_{i=1}^{n} \mathbb{1}_b(X_i) + Z_b \right) . \tag{3}$$

We indexed the $Z$'s by a bin instead of an integer without ambiguity. Note that by taking $Z$ almost-surely constant to 0, one recovers the usual (non-private) histogram estimator of a density.

### 3.1   General utility of histogram estimators

Characterizing the utility of (3) typically requires assumptions on the distribution $\pi$ to estimate. The class of $L$-Lipschitz densities is defined as

$$\Theta_L^{\text{Lip}} := \left\{ \pi \in \mathcal{C}^0([0,1], \mathbb{R}_+) \,\middle|\, \begin{cases} \forall x, y \in [0,1], |\pi(y) - \pi(x)| \le L|y-x| , \\ \int_{[0,1]} \pi = 1 . \end{cases} \right\} . \tag{4}$$

The following general-purpose lemma gives an upper-bound on the error that the histogram estimator makes on Lipschitz distributions:

**Lemma 1** (General utility of (3)). *There exists $C_L > 0$, a positive constant that only depends on $L$, such that*

$$\sup_{x_0 \in [0,1]} \sup_{\pi \in \Theta_L^{\text{Lip}}} \mathbb{E}_{\mathbf{X} \sim \mathbb{P}_\pi^{\otimes n}, \hat{\pi}^{\text{hist}}} \left( \left( \hat{\pi}^{\text{hist}}(\mathbf{X})(x_0) - \pi(x_0) \right)^2 \right) \le C_L \left( h^2 + \frac{1}{nh} + \frac{\mathbb{V}(Z)}{n^2 h^2} \right) .$$

The proof is given in Appendix C. The term $h^2$ corresponds to the bias of the estimator. The variance term $\frac{1}{nh} + \frac{\mathbb{V}(Z)}{n^2 h^2}$ exhibits two distinct contributions : the sampling noise $\frac{1}{nh}$ and the privacy noise $\frac{\mathbb{V}(Z)}{n^2 h^2}$. In particular, the utility of $\hat{\pi}^{\text{hist}}$ changes depending whether the variance is dominated by the sampling noise or by the privacy noise.

### 3.2   Privacy and bin size tuning

$\hat{\pi}^{\text{hist}}(\mathbf{X})$ is a simple function of the bin count vector $f(\mathbf{X}) := \left( \sum_{i=1}^n \mathbb{1}_{b_1}(X_i), \dots, \sum_{i=1}^n \mathbb{1}_{b_{1/h}}(X_i) \right)$. In particular, since the bins form a partition of $[0,1]$, changing the value of one of the $X$'s can change the values of at most two components of $f(\mathbf{X})$ by at most 1. Hence, the $l_1$ and $l_2$ sensitivities of $f$ are respectively 2 and $\sqrt{2}$. By a direct application of the Laplace or Gaussian mechanisms, and by choosing the binsize that minimizes the variance, we obtain the following privacy-utility result :

**Theorem 1** (Privacy and utility of (3) - DP case). *Given $\epsilon > 0$, using $\hat{\pi}^{\text{hist}}$ with $h = \max(n^{-1/3},$ $(n\epsilon)^{-1/2})$ and $Z = \frac{2}{\epsilon}\mathcal{L}(1)$, where $\mathcal{L}(1)$ refers to a random variable following a Laplace distribution of parameter 1, leads to an $\epsilon$-DP procedure. Furthermore, in this case, there exists $C_L > 0$, a positive constant that*

*only depends on L, such that*

$$\sup_{x_0 \in [0,1]} \sup_{\pi \in \Theta_L^{\mathrm{Lip}}} \mathbb{E}_{\mathbf{X} \sim \mathbb{P}_\pi^{\otimes n}, \hat{\pi}^{\mathrm{hist}}} \left( \left( \hat{\pi}^{\mathrm{hist}}(\mathbf{X})(x_0) - \pi(x_0) \right)^2 \right) \leq C_L \max\left\{ n^{-2/3}, (n\epsilon)^{-1} \right\} .$$

*Furthermore, given $\rho > 0$, using $\hat{\pi}^{hist}$ with $h = \max(n^{-1/3}, (n\sqrt{\rho})^{-1/2})$ and $Z = \sqrt{\frac{1}{\rho}} \mathcal{N}(0,1)$, where $\mathcal{N}(0,1)$ refers to a random variable following a centered Gaussian distribution of variance 1, leads to a $\rho$-zCDP procedure. Furthermore, in this case, there exists $C_L > 0$, a positive constant that only depends on $L$, such that*

$$\sup_{x_0 \in [0,1]} \sup_{\pi \in \Theta_L^{Lip}} \mathbb{E}_{\mathbf{X} \sim \mathbb{P}_\pi^{\otimes n}, \hat{\pi}^{hist}} \left( \left( \hat{\pi}^{hist}(\mathbf{X})(x_0) - \pi(x_0) \right)^2 \right) \leq C_L \max\left\{ n^{-2/3}, (n\sqrt{\rho})^{-1} \right\} .$$

Note that this bound is uniform in $x_0$, which is more general than the integrated upper-bounds presented in Barber & Duchi (2014). In particular, by integration on $[0,1]$, the same bound also holds for the integrated risk (in $L^2$ norm), which recovers the version of Barber & Duchi (2014). As expected, the optimal bin size $h$ depends on the sample size $n$ and on the parameter ($\epsilon$ or $\rho$) tuning the privacy. Also note that $\rho$-zCDP version may also be obtained by the relations between $\epsilon$-DP and $\rho$-zCDP (see Bun & Steinke (2016)).

### 3.3 Lower-bounds and minimax optimality

All lower-bounds will be investigated in a minimax sense. Given a class $\Pi$ of admissible densities, a semi-norm $\|\cdot\|$ on a space containing the class $\Pi$, and a non-decreasing positive function $\Phi$ such that $\Phi(0) = 0$, the minimax risk is defined as

$$\inf_{\hat{\pi} \text{ s.t. } \mathcal{C}} \sup_{\pi \in \Pi} \mathbb{E}_{\mathbf{X} \sim \mathbb{P}_\pi^{\otimes n}, \hat{\pi}} \Phi(\|\hat{\pi}(\mathbf{X}) - \pi\|) ,$$

where $\mathcal{C}$ is a condition that must satisfy the estimator (privacy in our case).

**General framework.** A usual technique for the derivation of minimax lower bounds on the risk uses a reduction to a testing problem (see Tsybakov (2009)). Indeed, if a family $\Pi' := \{\pi_1, \ldots, \pi_m\} \subset \Pi$ of cardinal $m$ is an $\Omega$-packing of $\Pi$ (that is if $i \neq j \implies \|\pi_i - \pi_j\| \geq 2\Omega$), then a lower bound is given by

$$\inf_{\hat{\pi} \text{ s.t. } \mathcal{C}} \sup_{\pi \in \Pi} \mathbb{E}_{\mathbf{X} \sim \mathbb{P}_\pi^{\otimes n}, \hat{\pi}} \Phi(\|\hat{\pi}(\mathbf{X}) - \pi\|)$$
$$\geq \Phi(\Omega) \inf_{\substack{\hat{\pi} \text{ s.t. } \mathcal{C} \\ \Psi : \mathrm{codom}(\hat{\pi}) \to \{1, \ldots, m\}}} \max_{i \in \{1, \ldots, m\}} \mathbb{P}_{\mathbf{X} \sim \mathbb{P}_{\pi_i}^{\otimes n}, \hat{\pi}} \left( \Psi\left(\hat{\pi}(\mathbf{X})\right) \neq i \right) . \tag{5}$$

For more details, see Duchi et al. (2016); Acharya et al. (2021e); Lalanne et al. (2023a). The right-hand side characterizes the difficulty of discriminating the distributions of the packing by a statistical test. Independently on the condition $\mathcal{C}$, it can be lower-bounded using information-theoretic results such a Le Cam's lemma (Rigollet & Hütter, 2015, Lemma 5.3) or Fano's lemma (Giraud, 2021, Theorem 3.1). When $\mathcal{C}$ is a local privacy condition, Duchi et al. (2016) provides analogous results that take privacy into account. Recent work (Acharya et al., 2021e; Kamath et al., 2022; Lalanne et al., 2023a) provides analogous forms for multiple notions of *central* privacy. When using this technique, finding good lower-bounds on the minimax risk boils down to finding a packing of densities that are far enough from one another without being too easy to discriminate with a statistical test.

It is interesting to note that for the considered problem, this technique does not yield satisfying lower-bounds with $\rho$-zCDP every time Fano's lemma is involved. Systematically, a small order is lost. To circumvent that difficulty, we had to adapt Assouad's technique to the context of $\rho$-zCDP. Similar ideas have been used in Duchi et al. (2016) for lower-bounds under *local* differential privacy and in Acharya et al. (2021e) for regular *central* differential privacy. To the best of our knowledge, such a technique has never been used in the context of central *concentrated* differential privacy, and is presented in Appendix D. In all the proofs of the lower-bounds, we systematically presented both approaches whenever there is a quantitative difference. This difference could be due to small suboptimalities in Fano's lemma for concentrated differential privacy, or simply to the use of a suboptimal packing.

### 3.3.1 Pointwise lower-bound

The first lower-bound that will be investigated is with respect to the pointwise risk. Pointwise, that is to say given $x_0 \in [0, 1]$, the performance of the estimator $\hat{\pi}$ is measured by how well it approximates $\pi$ at $x_0$ with the quadratic risk $\mathbb{E}_{\mathbf{X} \sim \mathbb{P}_{\pi}^{\otimes n}, \hat{\pi}} \left( ((\hat{\pi}(\mathbf{X})(x_0) - \pi(x_0))^2 \right)$. Technically, it is the easiest since it requires a "packing" of only two elements, which gives the following lower-bound:

**Theorem 2** (Pointwise lower-bound). *There exists $C_L > 0$, a positive constant depending only on $L$ such that, for any $x_0 \in [0, 1]$, there exist $n_0(x_0, L) \in \mathbb{N}$ and $c_0(x_0, L) > 0$ such that for any $n \geq n_0$, and any $\alpha \geq c_0/n$*

$$\inf_{\hat{\pi} \text{ s.t.} \mathcal{C}} \sup_{\pi \in \Theta_L^{\text{Lip}}} \mathbb{E}_{\mathbf{X} \sim \mathbb{P}_{\pi}^{\otimes n}, \hat{\pi}} \left( ((\hat{\pi}(\mathbf{X})(x_0) - \pi(x_0))^2 \right) \geq C_L^{-1} \max \left\{ n^{-2/3}, (n\alpha)^{-1} \right\} , \tag{6}$$

*where $\alpha = \epsilon$ when the condition $\mathcal{C}$ is the $\epsilon$-DP condition and $\alpha = \sqrt{\rho}$ when $\mathcal{C}$ is $\rho$-zCDP.*

*Proof idea.* Let $x_0 \in [0, 1]$. As explained above, finding a "good" lower-bound can be done by finding and analyzing a "good" packing of the parameter space. Namely, in this case, we have to find distributions on $[0, 1]$ that have a $L$-Lipschitz density (w.r.t. Lebesgue's measure) such that the densities are far from one another at $x_0$, but such that it is not extremely easy to discriminate them with a statistical test. We propose to use a packing $\{\mathbb{P}_f, \mathbb{P}_g\}$ of two elements where $g$ is the constant function on $[0, 1]$ (hence $\mathbb{P}_g$ is the uniform distribution) and $f$ deviates from $g$ by a small triangle centered at $x_0$. The two densities are represented in Figure 1.

After analyzing various quantities about these densities, such as their distance at $x_0$, their KL divergences or their TV distance, we leverage Le Cam-type results to conclude. $\square$

The full proof can be found in Appendix E.

Additionally, we can notice that, when applied to any fixed $x_0 \in [0, 1]$, Theorem 2 immediately gives the following corollary for the control in infinite norm :

**Corollary 1** (Infinite norm lower-bound). *There exists $C_L > 0$, a positive constant depending only on $L$ such that there exist $n_0(L) \in \mathbb{N}$ and $c_0(L) > 0$ such that for any $n \geq n_0$, and any $\alpha \geq c_0/n$*

$$\inf_{\hat{\pi} \text{ s.t.} \mathcal{C}} \sup_{\pi \in \Theta_L^{\text{Lip}}} \mathbb{E}_{\mathbf{X} \sim \mathbb{P}_{\pi}^{\otimes n}, \hat{\pi}} \|\hat{\pi}(\mathbf{X}) - \pi\|_{\infty}^2 \geq C_L^{-1} \max \left\{ n^{-2/3}, (n\alpha)^{-1} \right\} , \tag{7}$$

*where $\alpha = \epsilon$ when the condition $\mathcal{C}$ is the $\epsilon$-DP condition and $\alpha = \sqrt{\rho}$ when $\mathcal{C}$ is $\rho$-zCDP.*

**On the optimality and on the cost of privacy.** Theorem 1, Theorem 2 and Corollary 1 give the following general result : Under $\epsilon$-DP or under $\rho$-zCDP, histogram estimators have minimax-optimal rates of convergence against distributions with Lipschitz densities, for the pointwise risk or the risk in infinite norm. In particular, in the *low privacy* regime ("large" $\alpha$), the usual minimax rate of estimation of $n^{-\frac{2}{3}}$ is not degraded. This includes the early observations of Wasserman & Zhou (2010) in the case of constant $\alpha$ ($\epsilon$ or $\sqrt{\rho}$). However, in the *high privacy* regimes ($\alpha \ll n^{-\frac{1}{3}}$), these results prove a systematic degradation of the estimation. Those regimes are the same as in Barber & Duchi (2014), the metrics on the other hand are different.

### 3.3.2 Integrated lower-bound

The lower-bound of Theorem 2 is interesting, but its pointwise (or in infinite norm in the case of Corollary 1) nature means that much global information is possibly lost. Instead, one can look at the integrated risk $\mathbb{E}_{\mathbf{X} \sim \mathbb{P}_{\pi}^{\otimes n}, \hat{\pi}} \|\hat{\pi}(\mathbf{X}) - \pi\|_{L^2}^2$. Given Lemma 1 and the fact that we work on probability distributions with a compact support, upper-bounding this quantity is straightforward.

The lower-bound for the integrated risk is given by :

**Theorem 3** (Integrated lower-bound). *There exists $C_L > 0$, a positive constant depending only on $L$ such that, there exist $n_0(L) \in \mathbb{N}$ and $c_0(L) > 0$ such that for any $n \geq n_0$, and any $\alpha \geq c_0/n$*

$$\inf_{\hat{\pi} \ s.t. \ \mathcal{C}} \sup_{\pi \in \Theta_L^{Lip}} \mathbb{E}_{\mathbf{X} \sim \mathbb{P}_\pi^{\otimes n}, \hat{\pi}} \|\hat{\pi}(\mathbf{X}) - \pi\|_{L^2}^2 \geq C_L^{-1} \max \left\{ n^{-2/3}, (n\alpha)^{-1} \right\}$$

*where $\alpha = \epsilon$ when $\mathcal{C}$ is the $\epsilon$-DP condition, and $\alpha = \sqrt{\rho}$ when $\mathcal{C}$ is the $\rho$-zCDP condition.*

*Proof idea.* If we were to use the same packing (see Figure 1) as in the proof of Theorem 2, the lower-bounds would not be good. Indeed, moving from the pointwise difference to the $L^2$ norm significantly diminishes the distances in the packing. Instead, we will use the same idea of deviating from a constant function by triangles, except that we authorize more than one deviation. More specifically, we consider a packing consisting of densities $f_\omega$'s where the $\omega$'s are a well-chosen family of $\{0,1\}^m$ ($m$ is fixed in the proof) (Van der Vaart, 1998). Then, for a given $\omega \in \{0,1\}^m$, $f_\omega$ has a triangle centered on $\frac{i}{m+1}$ iff $w_i \neq 0$.

We then leverage Fano-type inequalities, and we use Assouad's method in order to find the announced lower-bounds. □

The full proof is in Appendix F.

Since the lower-bounds of Theorem 3 match the upper-bounds of Theorem 1, we conclude that the corresponding estimators are optimal in terms of minimax rate of convergence.

## 4 Projection Estimators and Periodic Sobolev Densities

The Lipschitz densities considered in Section 3 are general enough to be applicable in many problems. However, this level of generality becomes a curse in terms of rate of estimation. Indeed, as we have seen, the optimal rate of estimation is $\max\left(n^{-2/3}, (n\epsilon)^{-1}\right)$. To put it into perspective, for many parametric estimation procedures, the optimal rate of convergence usually scales as $\max\left(n^{-1}, (n\epsilon)^{-2}\right)$ (Acharya et al., 2021e). This section studies the estimation of smoother distributions, for different smoothness levels, at the cost of generality. In particular, it establishes that the smoother the distribution class is, the closer the private rate of estimation is to $\max\left(n^{-1}, (n\epsilon)^{-2}\right)$. In other words, it means that the more regular the density is supposed to be, the closer we get to the difficulty of parametric estimation.

When the density of interest $\pi$ is in $L^2([0,1])$, it is possible to approximate it by projections. Indeed, $L^2([0,1])$ being a separable Hilbert space, there exists a countable orthonormal family $(\phi_i)_{i \in \mathbb{N} \setminus \{0\}}$ that is a Hilbert basis. In particular, if $\theta_i := \int_{[0,1]} \pi \, \phi_i$ then

$$\sum_{i=1}^N \theta_i \phi_i \xrightarrow[N \to +\infty]{L^2} \pi \ .$$

Let $N$ be a positive integer, $Z_1, \ldots, Z_N$ be independent and identically distributed random variables with the same distribution as a centered random variable $Z$ having a finite variance. Given a dataset $\mathbf{X} = (X_1, \ldots, X_n)$, that is also independent of $Z_1, \ldots, Z_N$, the (randomized) projection estimator is defined as

$$\hat{\pi}^{\text{proj}}(\mathbf{X}) = \sum_{i=1}^N \left( \hat{\theta}_i + \frac{1}{n} Z_i \right) \phi_i \quad \text{where} \quad \hat{\theta}_i := \frac{1}{n} \sum_{j=1}^n \phi_i(X_j) \ . \tag{8}$$

The truncation order $N$ and the random variable $Z$ are tuned later to obtain the desired levels of privacy and utility. $L^2([0,1])$ has many well known Hilbert bases, hence multiple choices for the family $(\phi_i)_{i \in \mathbb{N} \setminus \{0\}}$. For instance, orthogonal polynomials, wavelets, or the Fourier basis, are often great choices for projection estimators. Because of the privacy constraint however, it is better to consider a *uniformly bounded* Hilbert basis (Wasserman & Zhou, 2010), which is typically not the case with a polynomial or wavelet basis. From now on, this work will focus on the following Fourier basis :

$$\phi_1(x) = 1$$
$$\phi_{2k}(x) = \sqrt{2}\sin(2\pi kx) \quad k \geq 1$$
$$\phi_{2k+1}(x) = \sqrt{2}\cos(2\pi kx) \quad k \geq 1 \ .$$

Note that we used the *upper* notation $\pi$ to refer to the real number $3, 14\ldots$, which is not to be mistaken for the *lower* notation $\pi$, which is reserved for the density of the distribution of interest. This shouldn't introduce any ambiguity since $\pi$ is only used locally when looking at Fourier coefficients, and is often simply hidden in the constants.

### 4.1 General utility of projection estimators

By the Parseval formula, the truncation resulting of approximating the density $\pi$ on a finite family of $N$ orthonormal functions induces a bias term that accounts for $\sum_{i \geq N+1} \theta_i^2$ in the mean square error. Characterizing the utility of $\hat{\pi}^{\mathrm{proj}}$ requires controlling this term, and this is usually done by imposing that $\pi$ is in a Sobolev space. We recall the definition given in Tsybakov (2009): given $\beta \in \mathbb{N} \setminus \{0\}$ and $L > 0$, the class $\Theta_{L,\beta}^{\mathrm{Sob}}$ of Sobolev densities of parameters $\beta$ and $L$ is defined as

$$\Theta_{L,\beta}^{\mathrm{Sob}} := \left\{ \pi \in \mathcal{C}^\beta([0,1], \mathbb{R}_+) \,\middle|\, \begin{cases} \pi^{(\beta-1)} \text{ is absolutely continuous}, \\ \int_{[0,1]} \left(\pi^{(\beta)}\right)^2 \leq L^2, \\ \int_{[0,1]} \pi = 1. \end{cases} \right\}.$$

For a function $f$, we used the notation $f^{(\beta)}$ to refer to its derivative of order $\beta$. In addition, the class $\Theta_{L,\beta}^{\mathrm{PSob}}$ of periodic Sobolev densities of parameters $\beta$ and $L$ is defined as

$$\Theta_{L,\beta}^{\mathrm{PSob}} := \left\{ \pi \in \Theta_{L,\beta}^{\mathrm{Sob}} \,\middle|\, \forall j \in \{0, \ldots, \beta-1\}, \pi^{(j)}(0) = \pi^{(j)}(1) \right\}. \tag{9}$$

Finally, we recall the following general-purpose lemma (Tsybakov, 2009) that allows controlling the truncation bias :

**Fact 1** (Ellipsoid reformulation (Tsybakov, 2009)). *A non-negative function $\pi$ with integral 1 belongs to* $\Theta_{L,\beta}^{PSob}$ *if and only if* $\sum_{i=1}^{\infty} a_i^{2\beta} \theta_i^2 \leq \dfrac{L^2}{\pi^{2\beta}}$, *where* $a_j := j$ *if $j$ is even and* $a_j := j - 1$ *if $j$ is odd.*

In this class, one can characterize the utility of projection estimators with the following lemma:

**Lemma 2** (General utility of (8)). *There is a constant $C_{L,\beta} > 0$, depending only on $L, \beta$, such that*

$$\sup_{\pi \in \Theta_{L,\beta}^{PSob}} \mathbb{E}_{\mathbf{X} \sim \mathbb{P}_\pi^{\otimes n}, \hat{\pi}^{proj}} \|\hat{\pi}^{proj}(\mathbf{X}) - \pi\|_{L^2}^2 \leq C_{L,\beta} \left( \frac{1}{N^{2\beta}} + \frac{N}{n} + \frac{N\mathbb{V}(Z)}{n^2} \right).$$

The proof can be found in Appendix G

### 4.2 Privacy and bias tuning

The estimator $\hat{\pi}^{\mathrm{proj}}(\mathbf{X})$ is a function of the sums $\left( \sum_{j=1}^n \phi_1(X_j), \ldots, \sum_{j=1}^n \phi_N(X_j) \right)$. In particular, it is possible to use Laplace and Gaussian mechanisms on this function in order to obtain privacy. Since the functions $|\phi_i|$ are bounded by $\sqrt{2}$ for any $i$, the $l_1$ sensitivity of this function is $2\sqrt{2}N$ and its $l_2$ sensitivity is $2\sqrt{2}\sqrt{N}$. Applying the Laplace and the Gaussian mechanism and tuning $N$ to optimize the utility of Lemma 2 gives the following result:

**Theorem 4** (Privacy and utility of (8)). *Given any $\epsilon > 0$ and truncation order $N$, using $\hat{\pi}^{proj}$ with $Z = \frac{2N\sqrt{2}}{\epsilon}\mathcal{L}(1)$, where $\mathcal{L}(1)$ refers to a random variable following a Laplace distribution of parameter 1, leads to an $\epsilon$-DP procedure. Moreover, there exists $C_{L,\beta} > 0$, a positive constant that only depends on $L$ and $\beta$, such that if $N$ is of the order of $\min \left( n^{\frac{1}{2\beta+1}}, (n\epsilon)^{\frac{1}{\beta+3/2}} \right)$,*

$$\sup_{\pi \in \Theta_{L,\beta}^{PSob}} \mathbb{E}_{\mathbf{X} \sim \mathbb{P}_\pi^{\otimes n}, \hat{\pi}^{proj}} \|\hat{\pi}^{proj}(\mathbf{X}) - \pi\|_{L^2}^2 \leq C_{L,\beta} \max \left\{ n^{-\frac{2\beta}{2\beta+1}}, (n\epsilon)^{-\frac{2\beta}{\beta+3/2}} \right\}.$$

*Furthermore, given any $\rho > 0$, and truncation order $N$, using $\hat{\pi}^{proj}$ with $Z = \frac{2\sqrt{N}}{\sqrt{\rho}}\mathcal{N}(0,1)$, where $\mathcal{N}(0,1)$ refers to a random variable following a centered Gaussian distribution of variance 1, leads to a $\rho$-zCDP*

*procedure. Moreover, there exists $C_{L,\beta} > 0$, a positive constant that only depends on $L$ and $\beta$, such that, if $N$ is of the order of* $\min\left(n^{\frac{1}{2\beta+1}}, \left(n\sqrt{\rho}\right)^{\frac{1}{\beta+1}}\right)$

$$\sup_{\pi \in \Theta^{PSob}_{L,\beta}} \mathbb{E}_{\mathbf{X} \sim \mathbb{P}_\pi^{\otimes n}, \hat{\pi}^{proj}} \|\hat{\pi}^{proj}(\mathbf{X}) - \pi\|^2_{L^2} \leq C_{L,\beta} \max\left\{n^{-\frac{2\beta}{2\beta+1}}, \left(n\sqrt{\rho}\right)^{-\frac{2\beta}{\beta+1}}\right\} \ .$$

We now discuss these guarantees depending on the considered privacy regime.

**Low privacy regimes.** According to Theorem 4, when the privacy-tuning parameters are not too small (i.e. when the estimation is not too private), the usual rate of convergence $n^{-\frac{2\beta}{2\beta+1}}$ is not degraded. In particular, for constant $\epsilon$ or $\rho$, this recovers the results of Wasserman & Zhou (2010).

**High privacy regimes.** Furthermore, Theorem 4 tells that in high privacy regimes ($\epsilon \ll n^{-\frac{\beta-1/2}{2\beta+1}}$ or $\rho \ll n^{-\frac{2\beta+2}{2\beta+1}}$), the provable guarantees of the projection estimator are degraded compared to the usual rate of convergence. Is this degradation constitutive of the estimation problem, or is it due to a suboptimal upper-bound? Section 4.3 shows that this excess of risk is in fact almost optimal.

### 4.3 Lower-bounds

As with the integrated risk on Lipschitz distributions, obtaining lower-bounds for the class of periodic Sobolev densities is done by considering a packing with many elements. The idea of the packing is globally the same as for histograms, except that the uniform density is perturbed with a general $C^\infty$ kernel with compact support instead of simple triangles. In the end, we obtain the following result:

**Theorem 5** (Integrated lower-bound). *Given $L, \beta > 0$ there exists constants $C_{L,\beta} > 0$, $n_0(L,\beta) \in \mathbb{N}$, and $c_0(L,\beta) > 0$, such that for any $n \geq n_0$, and any $\alpha \geq c_0/n$*

$$\inf_{\hat{\pi} \ s.t. \ \mathcal{C}} \sup_{\pi \in \Theta^{PSob}_{L,\beta}} \mathbb{E}_{\mathbf{X} \sim \mathbb{P}_\pi^{\otimes n}, \hat{\pi}} \|\hat{\pi}(\mathbf{X}) - \pi\|^2_{L^2} \geq C_{L,\beta}^{-1} \max\left\{n^{-\frac{2\beta}{2\beta+1}}, (n\alpha)^{-\frac{2\beta}{\beta+1}}\right\}$$

*where $\alpha = \epsilon$ when $\mathcal{C}$ is the $\epsilon$-DP condition, and $\alpha = \sqrt{\rho}$ when $\mathcal{C}$ is the $\rho$-zCDP condition.*

*Proof idea.* As with the proof of Theorem 3, this lower-bound is based on the construction of a packing of densities $f_\omega$'s where the $\omega$'s are a well-chosen family of $\{0,1\}^m$ ($m$ is fixed in the proof). Then, for a given $\omega \in \{0,1\}^m$, $f_\omega$ deviates from a constant function around $\frac{i}{m+1}$ if, and only if, $w_i \neq 0$. Contrary to the proof of Theorem 3 however, the deviation cannot be by a triangle : Indeed, such a function wouldn't even be differentiable. Instead, we use a deviation by a $C^\infty$ kernel with compact support. Even if the complete details are given in the full proof, Figure 3 gives a general illustration of the packing.

Again, Fano-type inequalities (for the $\epsilon$-DP case), and Assouad's lemma (for the $\rho$-zCDP case) are used to conclude. $\qquad\square$

The full proof can be found in Appendix H. In comparison with the upper-bounds of Theorem 4, for $\epsilon$-DP the lower-bound *almost* matches the guarantees of the projection estimator. In particular, the excess of risk in the high privacy regime is close to being optimal. Section 4.4 explains how to bridge the gap even more, at the cost of relaxation.

Under $\rho$-zCDP, the lower-bounds and upper-bounds actually match. We conclude that projection estimators with $\rho$-zCDP obtain minimax-optimal rates of convergence.

### 4.4 Near minimax optimality via relaxation

An hypothesis that we can make on the sub-optimality of the projection estimator against $\epsilon$-DP mechanisms is that the $l_1$ sensitivity of the estimation of $N$ Fourier coefficients scales as $N$ whereas its $l_2$ sensitivity scales as $\sqrt{N}$. Traditionally, the Gaussian mechanism (Dwork et al., 2006a;b) has allowed to use the $l_2$ sensitivity instead of the $l_1$ one at the cost of introducing a relaxation term $\delta$ in the privacy guarantees, leading to $(\epsilon, \delta)$-DP.

A direct application of the Gaussian mechanism Dwork & Roth (2014) thus tells that $\hat{\pi}^{\mathrm{proj}}$ with $Z = \frac{4\sqrt{\ln{(1.25/\delta)}}\sqrt{N}}{\epsilon}\mathcal{N}(0,1)$ is $(\epsilon,\delta)$-DP and, by Lemma 2, has an error bounded as

$$\sup_{\pi\in\Theta_{L,\beta}^{\mathrm{PSob}}} \mathbb{E}_{\mathbf{X}\sim\mathbb{P}_{\pi}^{\otimes n},\hat{\pi}^{\mathrm{proj}}}\|\hat{\pi}^{\mathrm{proj}}(\mathbf{X}) - \pi\|_{L^2}^2 \leq C_{L,\beta}\left(\frac{1}{N^{2\beta}} + \frac{N}{n} + \frac{16N^2\ln{(1.25/\delta)}}{\epsilon^2 n^2}\right) \ .$$

Thus, choosing $N$ of the order of $\min\left(n^{\frac{1}{2\beta+1}}, \left(\frac{n\epsilon}{\sqrt{\ln(1.25/\delta)}}\right)^{\frac{1}{\beta+1}}\right)$ leads to a general error as

$$\sup_{\pi\in\Theta_{L,\beta}^{\mathrm{PSob}}} \mathbb{E}_{\mathbf{X}\sim\mathbb{P}_{\pi}^{\otimes n},\hat{\pi}^{\mathrm{proj}}}\|\hat{\pi}^{\mathrm{proj}}(\mathbf{X}) - \pi\|_{L^2}^2 \leq C_{L,\beta}\max\left\{n^{-\frac{2\beta}{2\beta+1}}, \left(\frac{n\epsilon}{\sqrt{\ln{(1.25/\delta)}}}\right)^{-\frac{2\beta}{\beta+1}}\right\} \ .$$

Finally, it can be interesting to look at prescribed rates for $\delta$ as a function of $n$.

**Corollary 2** (Privacy and utility of (8) with relaxation). *Consider $\gamma > 0$, $n$ and integer, and $0 < \epsilon \leq 8\ln n^{\gamma}$. Defining $\tilde{\rho} := \frac{1}{16}\frac{\epsilon^2}{\ln(n^{\gamma})}$ and using $\hat{\pi}^{proj}$ with $Z = \frac{2\sqrt{N}}{\sqrt{\tilde{\rho}}}\mathcal{N}(0,1)$, where $\mathcal{N}(0,1)$ refers to a random variable following a centered Gaussian distribution of variance 1, leads to an $\left(\epsilon, \frac{1}{n^{\gamma}}\right)$-DP procedure. there exists $C_{L,\beta} > 0$, a positive constant that only depends on $L$ and $\beta$, such that if $N$ is of the order of $\min\left(n^{\frac{1}{2\beta+1}}, \left(\frac{n}{\sqrt{\ln n}}\cdot\frac{\epsilon}{\sqrt{\gamma}}\right)^{\frac{1}{\beta+1}}\right)$ then*

$$\sup_{\pi\in\Theta_{L,\beta}^{PSob}} \mathbb{E}_{\mathbf{X}\sim\mathbb{P}_{\pi}^{\otimes n},\hat{\pi}^{proj}}\|\hat{\pi}^{proj}(\mathbf{X}) - \pi\|_{L^2}^2 \leq C_{L,\beta}\max\left\{n^{-\frac{2\beta}{2\beta+1}}, P_{\beta,\gamma}(\ln(n))(n\epsilon)^{-\frac{2\beta}{\beta+1}}\right\} \ ,$$

*where $P_{\beta,\gamma}$ is a polynomial expression depending on $\beta$ and $\gamma$.*

*Proof.* Since $\epsilon \leq 8\ln(n^{\gamma})$, we have $\tilde{\rho} \leq 2\sqrt{\tilde{\rho}\ln(n^{\gamma})}$. By Theorem 4 the mechanism is $\tilde{\rho}$-zCDP, and satisfies the claimed upper bounds for $N$ on the considered order. By Bun & Steinke (2016) (that states that if a mechanism $\mathfrak{M}$ is $\rho$-zCDP, then it is $\left(\rho + 2\sqrt{\rho\ln(1/\delta)}, \delta\right)$-DP for any $\delta > 0$) it is thus $\left(4\sqrt{\tilde{\rho}\ln(n^{\gamma})}, \frac{1}{n^{\gamma}}\right)$-DP. $\qquad\square$

In order to understand the implications of this result, one must understand the role of $\delta$ in $(\epsilon,\delta)$-differential privacy. It is usually interpreted as the probability of the procedure not respecting the $\epsilon$-DP condition (Dwork & Roth, 2014). Hence, with probability $\delta$, the result is not guaranteed to be private. A general rule of thumb for choosing $\delta$ is to take it much smaller than $1/n$ so that each individual of the database only has a small chance of seeing its data leak (Dwork & Roth, 2014). Choosing $\delta = 1/n^{\gamma}$ for $\gamma > 1$ is hence considered a good choice for $\delta$.

With this relaxation, the upper-bound of Corollary 2 matches the lower-bound of Theorem 5 for $\epsilon$-DP up to polylog factors.

## 5 Conclusion

As we have seen throughout this article, under central privacy, one can usually distinguish two estimation regimes. In the *low* privacy regime, on the one hand, the estimation rate is not degraded compared to its non-private counterpart. This notably covers the early observation of Wasserman & Zhou (2010) for constant privacy budget. In the *high* privacy regime on the other hand, a provable degradation is unavoidable, and we extended the study of such regimes beyond the cases covered in Barber & Duchi (2014).

Besides examples in which the estimation is sharp in both regimes, we also presented some example in which there are *small* gaps between the proved upper-bounds and lower-bounds. These gaps are nevertheless very small, especially for high degrees of smoothness, and they can be bridged up to logarithmic factors with a *reasonable* and quite standard relaxation.

**Acknowledgement**

Aurélien Garivier acknowledges the support of the Project IDEXLYON of the University of Lyon, in the framework of the Programme Investissements d'Avenir (ANR-16-IDEX-0005), and Chaire SeqALO (ANR-20-CHIA-0020-01). This project was supported in part by the AllegroAssai ANR project ANR-19-CHIA-0009. Additionally, we thank the anonymous reviewers for their precious inputs and suggestions.

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

# A    Useful results from the litterature

**Fact 2** (Neyman-Pearson & Le Cam's lemma (Rigollet & Hütter, 2015, Lemma 5.3)). *Let $\mathbb{P}_1$, $\mathbb{P}_2$ be two probability distributions on a measure space $\mathcal{E}$, then*

$$
\inf_{\Psi:\mathcal{E}\to\{1,2\}} \max_{i\in\{1,2\}} \mathbb{P}_{\mathbf{X}\sim\mathbb{P}_i}\left(\Psi\left(\mathbf{X}\right)\neq i\right) \geq \frac{1}{2} \inf_{\Psi:\mathcal{E}\to\{1,2\}} \sum_{i=1}^{2} \mathbb{P}_{\mathbf{X}\sim\mathbb{P}_i}\left(\Psi\left(\mathbf{X}\right)\neq i\right)
$$
$$
= \frac{1}{2}\left(1 - \mathrm{TV}\left(\mathbb{P}_1,\mathbb{P}_2\right)\right) . \tag{10}
$$

**Fact 3** (Fano's lemma (Giraud, 2021, Theorem 3.1)). *Let $(\mathbb{P}_i)_{i\in\{1,\dots,N\}}$ be a family of probability distributions on a measure space $\mathcal{E}$. For any probability distribution $\mathbb{Q}$ on $\mathcal{E}$ such that $\mathbb{P}_i \ll \mathbb{Q}$ for all $i$, and for any test function $\Psi:\mathcal{X}^n \to \{1,\dots,N\}$,*

$$
\max_{i\in\{1,\dots,N\}} \mathbb{P}_{\mathbf{X}\sim\mathbb{P}_i}\left(\Psi\left(\mathbf{X}\right)\neq i\right) \geq \frac{1}{N}\sum_{i=1}^{N} \mathbb{P}_{\mathbf{X}\sim\mathbb{P}_i}\left(\Psi\left(\mathbf{X}\right)\neq i\right)
$$
$$
\geq 1 - \frac{1 + \frac{1}{N}\sum_{i=1}^{N}\mathrm{KL}\left(\mathbb{P}_i\|\mathbb{Q}\right)}{\ln(N)} . \tag{11}
$$

*Often $\mathbb{Q}$ is set to $\frac{1}{N}\sum_{i=1}^{N}\mathbb{P}_i$.*

**Fact 4** (Le Cam's lemma for differential privacy (Lalanne et al., 2023a, Theorem 1)). *If a randomized mechanism $\mathfrak{M}$ satisfies $(\epsilon,\delta)$-DP, then for any test function $\Psi:\mathrm{codom}\left(\mathfrak{M}\right)\to\{1,2\}$ and any probability distributions $\mathbb{P}_1$ and $\mathbb{P}_2$ on $\mathcal{X}$ we have*

$$
\max_{i\in\{1,2\}} \mathbb{P}_{\mathbf{X}\sim\mathbb{P}_i^{\otimes n},\mathfrak{M}}\left(\Psi\left(\mathfrak{M}\left(\mathbf{X}\right)\right)\neq i\right)
$$
$$
\geq \frac{1}{2}\left(\left(1 - \left(1-e^{-\epsilon}\right)\mathrm{TV}\left(\mathbb{P}_1,\mathbb{P}_2\right)\right)^n - 2ne^{-\epsilon}\delta\mathrm{TV}\left(\mathbb{P}_1,\mathbb{P}_2\right)\right) .
$$

**Fact 5.** *Le Cam's lemma for concentrated differential privacy (Lalanne et al., 2023a, Theorem 2)] If a randomized mechanism $\mathfrak{M}$ satisfies $\rho$-zCDP, then for any test function $\Psi:\mathrm{codom}\left(\mathfrak{M}\right)\to\{1,\dots,N\}$ and any probability distributions $\mathbb{P}_1$ and $\mathbb{P}_2$ on $\mathcal{X}$,*

$$
\max_{i\in\{1,2\}} \mathbb{P}_{\mathbf{X}\sim\mathbb{P}_i^{\otimes n},\mathfrak{M}}\left(\Psi\left(\mathfrak{M}\left(\mathbf{X}\right)\right)\neq i\right) \geq \frac{1}{2}\left(1 - n\sqrt{\rho/2}\mathrm{TV}\left(\mathbb{P}_1,\mathbb{P}_2\right)\right) .
$$

**Fact 6** (Fano's lemma for differential privacy (Lalanne et al., 2023a, Theorem 3)). *If a randomized mechanism $\mathfrak{M}$ satisfies $\epsilon$-DP, then for any test function $\Psi:\mathrm{codom}\left(\mathfrak{M}\right)\to\{1,\dots,N\}$, any family of probability distributions $(\mathbb{P}_i)_{i\in\{1,\dots,N\}}$ on $\mathcal{X}$,*

$$
\max_{i\in\{1,\dots,N\}} \mathbb{P}_{\mathbf{X}\sim\mathbb{P}_i^{\otimes n},\mathfrak{M}}\left(\Psi\left(\mathfrak{M}(\mathbf{X})\right)\neq i\right) \geq 1 - \frac{1 + \frac{n\epsilon}{N^2}\sum_{i,j}\frac{2\mathrm{TV}(\mathbb{P}_i,\mathbb{P}_j)}{1+\mathrm{TV}(\mathbb{P}_i,\mathbb{P}_j)}}{\ln(N)} .
$$

**Fact 7** (Fano's lemma for differential privacy (Lalanne et al., 2023a, Theorem 4)). *If a randomized mechanism $\mathfrak{M}$ satisfies $\epsilon$-DP, then for any test function $\Psi:\mathrm{codom}\left(\mathfrak{M}\right)\to\{1,\dots,N\}$, any family of probability distributions $(\mathbb{P}_i)_{i\in\{1,\dots,N\}}$ on $\mathcal{X}$,*

$$
\max_{i\in\{1,\dots,N\}} \mathbb{P}_{\mathbf{X}\sim\mathbb{P}_i^{\otimes n},\mathfrak{M}}\left(\Psi\left(\mathfrak{M}(\mathbf{X})\right)\neq i\right) \geq 1 - \frac{1 + \frac{n^2\rho}{N^2}\sum_{i,j}\frac{1}{n}\frac{2\mathrm{TV}(\mathbb{P}_i,\mathbb{P}_j)}{1+\mathrm{TV}(\mathbb{P}_i,\mathbb{P}_j)} + \left(\frac{2\mathrm{TV}(\mathbb{P}_i,\mathbb{P}_j)}{1+\mathrm{TV}(\mathbb{P}_i,\mathbb{P}_j)}\right)^2}{\ln(N)} .
$$

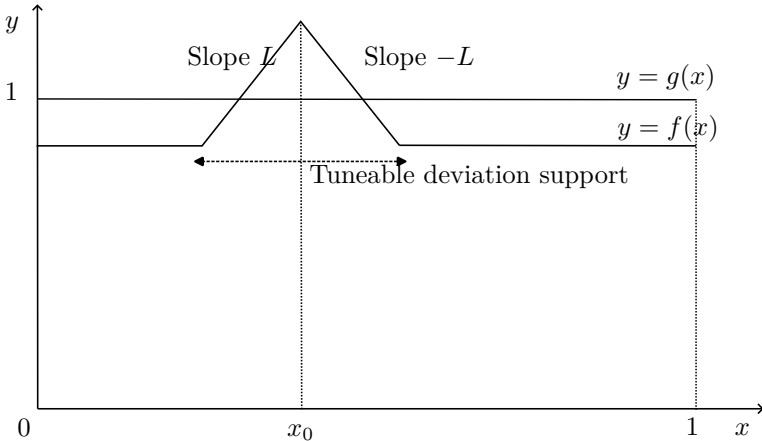

Figure 1: Packing for Theorem 2

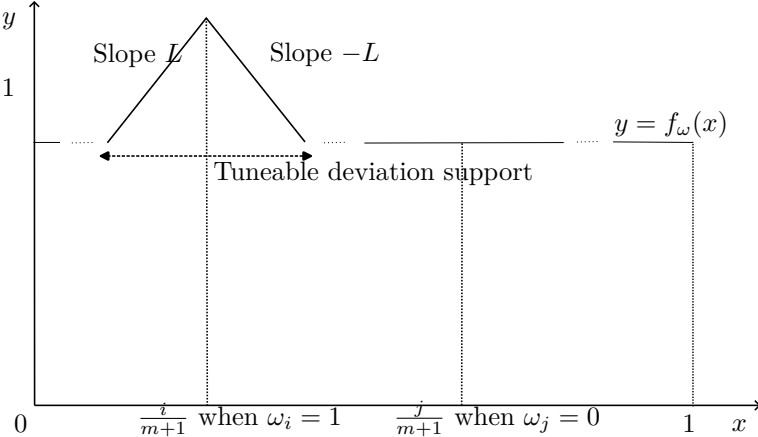

Figure 2: Packing for Theorem 3

## B Figures

## C Proof of Lemma 1

Let $\pi \in \Theta_L^{\text{Lip}}$, $x_0 \in [0, 1]$. The classical bias-variance decomposition gives that

$$\mathbb{E}\left(\left(\hat{\pi}^{\text{hist}}(\mathbf{X})(x_0) - \pi(x_0)\right)^2\right) = \left(\mathbb{E}\left(\hat{\pi}^{\text{hist}}(\mathbf{X})(x_0)\right) - \pi(x_0)\right)^2 + \mathbb{V}\left(\hat{\pi}^{\text{hist}}(\mathbf{X})(x_0)\right) .$$

For any $x \in [0, 1]$, we note $\text{bin}(x)$ the bin of the histogram in which $x$ falls into. Notice that, for any $x_0 \in [0, 1]$ and any integer $i$, the random variable $\mathbb{1}_{\text{bin}(x_0)}(X_i)$ follows a Bernoulli distribution of probability of success $\int_{\text{bin}(x_0)} \pi$. Let us first study the bias, using the definition (3) of $\hat{\pi}^{\text{hist}}$

$$\left|\mathbb{E}\left(\hat{\pi}^{\text{hist}}(\mathbf{X})(x_0)\right) - \pi(x_0)\right| = \left|\frac{1}{nh}\sum_{i=1}^{n}\mathbb{E}\left(\mathbb{1}_{\text{bin}(x_0)}(X_i)\right) - \pi(x_0)\right| = \left|\frac{n\int_{\text{bin}(x_0)}\pi(x)dx}{nh} - \pi(x_0)\right|$$

$$= \frac{1}{h}\left|\int_{\text{bin}(x_0)}(\pi(x) - \pi(x_0))dx\right| \leq \frac{1}{h}\int_{\text{bin}(x_0)}|\pi(x) - \pi(x_0)|\,dx \leq \frac{L}{h}\int_{\text{bin}(x_0)}|x - x_0|\,dx \leq \frac{Lh}{2} .$$

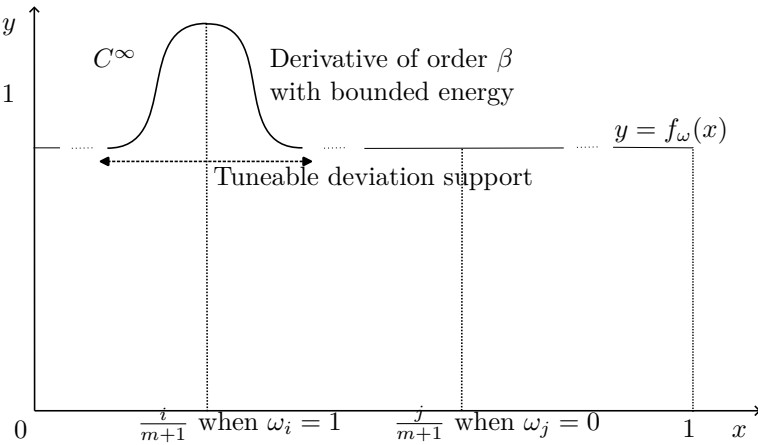

Figure 3: Packing for Theorem 5

Let us now look at the variance. By independence of $X_i$'s and $Z_j$'s,

$$
\mathbb{V}\left(\hat{\pi}^{\text{hist}}(\mathbf{X})(x_0)\right) = \frac{1}{n^2 h^2}\left(\sum_{i=1}^n \mathbb{V}\left(\mathbb{1}_{\text{bin}(x_0)}(X_i)\right) + \mathbb{V}\left(Z_{\text{bin}(x_0)}\right)\right)
$$

$$
= \frac{1}{n^2 h^2}\left(n\left(\int_{\text{bin}(x_0)}\pi\right)\left(1 - \int_{\text{bin}(x_0)}\pi\right) + \mathbb{V}(Z)\right)
$$

$$
\leq \frac{1}{nh^2}\left(\int_{\text{bin}(x_0)}\pi\right) + \frac{\mathbb{V}(Z)}{n^2 h^2}.
$$

Since $\pi$ is $L$-Lipschitz on $[0,1]$ and has to integrate to 1 (because it is a density), $\pi$ is uniformly bounded from above by $L+1$ on $[0,1]$. Hence, $\int_{\text{bin}(x_0)}\pi \leq (L+1)h$ and the result follows.

## D  Assouad's lemma with concentrated differential privacy.

As the reduction to a testing problem between multiple hypotheses, Assouad's lemma relies on similar ideas, where the packing has to be parametrized by a hypercube. Its advantage over tools like Fano's lemma is that it only makes tests between pairs of hypotheses (instead of all of them at the same time). The cost of this is that the control of the packing is slightly more difficult.

Suppose that the set of distributions of interest $\mathcal{P}$ contains a family of distributions $(\mathbb{P}_\omega)_{\omega \in \{0,1\}^m}$ for a certain positive integer $m$. If the loss function (taken quadratic for simplicity) can be decomposed as

$$
\forall \omega, \omega' \in \{0,1\}^m, \quad \|f_\omega - f_{\omega'}\|_{L^2}^2 \geq 2\tau \sum_{i=1}^m \mathbb{1}_{\omega_i \neq \omega_i'} = 2\tau d_{\text{ham}}(\omega, \omega') \ , \tag{12}
$$

where for any $\omega$, $f_\omega$ represents the density of $\mathbb{P}_\omega$, then the minimax risk can be lower-bounded as (the proof is classical and can be found in Acharya et al. (2021e, Section 5.4))

$$
\inf_{\hat{\pi} \text{ s.t. } \mathcal{C}} \sup_{\mathbb{P} \in \mathcal{P}} \mathbb{E}_{\mathbf{X} \sim \mathbb{P}, \hat{\pi}}\left(\|\hat{\pi}(\mathbf{X}) - \pi\|_{L^2}^2\right)
$$

$$
\geq \frac{\tau}{16}\sum_{i=1}^m \inf_{\substack{\mathfrak{M} \text{ s.t. } \mathcal{C} \\ \Psi:\text{codom}(\mathfrak{M})\to\{0,1\}}} \mathbb{P}_{\mathbf{X}\sim\mathbb{P}_{\omega^{i,0}}^{\otimes n},\mathfrak{M}}\left(\Psi\left(\mathfrak{M}(\mathbf{X})\right) \neq 0\right) + \mathbb{P}_{\mathbf{X}\sim\mathbb{P}_{\omega^{i,1}}^{\otimes n},\mathfrak{M}}\left(\Psi\left(\mathfrak{M}(\mathbf{X})\right) \neq 1\right) \ . \tag{13}
$$

where $\mathbb{P}_{\omega^{i,0}}$ and $\mathbb{P}_{\omega^{i,1}}$ are the *mixture* distributions

$$
\mathbb{P}_{\omega^{i,0}} := \frac{1}{2^{m-1}}\sum_{\omega \in \{0,1\}^m | \omega_i = 0} \mathbb{P}_\omega \quad \text{and} \quad \mathbb{P}_{\omega^{i,0}} := \frac{1}{2^{m-1}}\sum_{\omega \in \{0,1\}^m | \omega_i = 1} \mathbb{P}_\omega \ . \tag{14}
$$

The term

$$\mathbb{P}_{\mathbf{X}\sim\mathbb{P}_{\omega^{i,0}}^{\otimes n},\mathfrak{M}}\left(\Psi\left(\mathfrak{M}(\mathbf{X})\right)\neq 0\right)+\mathbb{P}_{\mathbf{X}\sim\mathbb{P}_{\omega^{i,1}}^{\otimes n},\mathfrak{M}}\left(\Psi\left(\mathfrak{M}(\mathbf{X})\right)\neq 1\right)$$

characterizes the *testing* difficulty between $\mathbb{P}_{\omega^{i,0}}$ and $\mathbb{P}_{\omega^{i,1}}$. It can be controlled by Le Cam's lemma, and by its variants when working under privacy (see Acharya et al. (2021e); Lalanne et al. (2023a) for differential privacy and Lalanne et al. (2023a) for concentrated differential privacy). Such results are reminded in Appendix A.

## E  Proof of Theorem 2

Let $x_0 \in (0,1)$. As explained in the sketch of the proof, we build a packing consisting of two elements, and after controlling quantities such as their KL divergences or their TV distances, we leverage Le Cam-type inequalities in order to obtain lower-bounds.

**Packing construction.**   We define the functions $f_{L,x_0,h}, \forall h > 0$ as

$$\forall x \in [0,1], \quad f_{L,x_0,h}(x) := \begin{cases} 1 - Lh^2 \text{ if } x \in [0, x_0 - h] \cup [x_0 + h, 1], \\ 1 - Lh^2 + Lh + L(x - x_0) \text{ if } x \in [x_0 - h, x_0) \,. \\ 1 - Lh^2 + Lh - L(x - x_0) \text{ if } x \in [x_0, x_0 + h) \end{cases} \tag{15}$$

Note that as soon as $h \leq \min\{x_0, 1 - x_0\}$, $f_{L,x_0,h} \in \Theta_L^{\mathrm{Lip}}$. The case $x_0 \in \{0, 1\}$ is treated in the exact same fashion, but by considering functions that only contain "half of a spike" centered on $x_0$. Furthermore, let us note $g$ the function that is constant to 1 on $[0, 1]$ (we have $g \in \Theta_L^{\mathrm{Lip}}$).

We start by recalling the total variation distance between two probability distributions, and we recall some useful alternative expressions that are used in the proofs of this article. Given $(\mathcal{U}, \mathcal{T})$ a set $\mathcal{U}$ equipped with a $\sigma$-algebra $\mathcal{T}$, and two probability measures $\mathbb{P}_1$ and $\mathbb{P}_2$ two probability distributions on $\mathcal{U}$, and compatible with $\mathcal{T}$, the total variation distance $\mathrm{TV}(\cdot, \cdot)$ between $\mathbb{P}_1$ and $\mathbb{P}_2$ is defined as

$$\mathrm{TV}\left(\mathbb{P}_1, \mathbb{P}_2\right) := \sup_{\mathcal{S} \in \mathcal{T}} \left|\mathbb{P}_1(\mathcal{S}) - \mathbb{P}_2(\mathcal{S})\right| \,.$$

Furthermore, when $\mathbb{P}_1, \mathbb{P}_2$ are dominated by a common $\sigma$-finite measure $\mu$ on $(\mathcal{U}, \mathcal{T})$, by noting $p_1 := \frac{d\mathbb{P}_1}{d\mu}$ and $p_2 := \frac{d\mathbb{P}_2}{d\mu}$, the Radon-Nikodym derivatives of $\mathbb{P}_1$ and $\mathbb{P}_2$ with respect to $\mu$, the following alternative expressions to the total variation can be useful :

$$\mathrm{TV}\left(\mathbb{P}_1, \mathbb{P}_2\right) := \sup_{\mathcal{S} \in \mathcal{T}} \left|\mathbb{P}_1(\mathcal{S}) - \mathbb{P}_2(\mathcal{S})\right| = \mathbb{P}_1(\{p_1 > p_2\}) - \mathbb{P}_2(\{p_1 > p_2\})$$
$$= \int_{\{p_1 > p_2\}} (p_1 - p_2)\,d\mu = \int_{\{p_2 \geq p_1\}} (p_2 - p_1)\,d\mu$$
$$= \frac{1}{2}\int_{\mathcal{U}} |p_1 - p_2|\,d\mu = 1 - \int_{\mathcal{U}} \min(p_1, p_2)\,d\mu \,.$$

These expressions simply come from considering the events $\{p_1 > p_2\}$ and $\{p_2 \geq p_1\}$ that form a partition of $\mathcal{U}$, and from the relation $|a - b| = a + b - 2\min(a, b)$ for any real numbers $a$ and $b$.

Jumping back to our original proof, when $f_{L,x_0,h} \in \Theta_L^{\mathrm{Lip}}$, we can compute the total variation between $\mathbb{P}_{f_{L,x_0,h}}$ and $\mathbb{P}_g$ the distributions of probability with densities $f_{L,x_0,h}$ and $g$ with respect to Lebesgue's measure on $[0, 1]$,

$$\mathrm{TV}\left(\mathbb{P}_{f_{L,x_0,h}}, \mathbb{P}_g\right) = 1 - \int_{[0,1]} \min\left(f_{L,x_0,h}, g\right) \overset{\overset{\text{Constant part}}{\leq}}{} 1 - \int_{[0,1]} 1 - Lh^2 dx = Lh^2 \,. \tag{16}$$

Another important measure of discrepancy between probability distributions is the so-called Kullback-Leibler (KL) divergence. For two probability distributions $\mathbb{P}$ and $\mathbb{Q}$ such that $\mathbb{P} \ll \mathbb{Q}$ (absolute continuity), it is defined as

$$\mathrm{KL}\left(\mathbb{P}\|\mathbb{Q}\right) = \int \ln\left(\frac{d\mathbb{P}}{d\mathbb{Q}}\right) d\mathbb{P} \,.$$

Back to our problem, for $h$ in a neighborhood of 0, we also have the following Taylor expansion on their KL divergence:

$$
\begin{aligned}
\mathrm{KL}\left(\mathbb{P}_g \,\|\, \mathbb{P}_{f_{L,x_0,h}}\right) &= \int_{[0,1]} \ln\left(\frac{g}{f_{L,x_0,h}}\right) g = \ln\left(\frac{1}{1-Lh^2}\right)(1-2h) + 2\int_0^h \ln\left(\frac{1}{1-Lh^2+Lt}\right) dt \\
&\leq C\left(h^3 + O(h^4)\right),
\end{aligned}
\tag{17}
$$

where $C$ is a positive constant depending only on $L$ the $O$ only hides constant factors. Furthermore, $|g(x_0) - f_{L,x_0,h}(x_0)| = L|h^2 - h|$ and $\{g, f_{L,x_0,h}\}$ is thus a $\frac{L}{2}|h^2 - h|$ packing of $\Theta_L^{\mathrm{Lip}}$ w.r.t the seminorm $f, g \mapsto \|f - g\| := |f(x_0) - g(x_0)|$.

**Recovering the usual lower-bound** By the classical minimax reduction as hypothesis testing Equation (5),

$$
\begin{aligned}
\inf_{\hat{\pi} \text{ s.t. } \mathcal{C}} &\sup_{\pi \in \Theta_L^{\mathrm{Lip}}} \mathbb{E}_{\mathbf{X} \sim \mathbb{P}_\pi^{\otimes n}, \hat{\pi}}\left(\left(\hat{\pi}(\mathbf{X})(x_0) - \pi(x_0)\right)^2\right) \\
&\geq \frac{L^2}{4}\left(h^2 - h\right)^2 \inf_{\hat{\pi} \text{ s.t. } \mathcal{C}} \inf_{\Psi:\Theta_L^{\mathrm{Lip}} \to \{0,1\}} \max\left\{\mathbb{P}_{\mathbf{X} \sim \mathbb{P}_g^{\otimes n}, \hat{\pi}}\left(\Psi(\hat{\pi}(\mathbf{X})) \neq 0\right),\right. \\
&\hspace{8cm} \left. \mathbb{P}_{\mathbf{X} \sim \mathbb{P}_{f_{L,x_0,h}}^{\otimes n}, \hat{\pi}}\left(\Psi(\hat{\pi}(\mathbf{X})) \neq 1\right)\right\} \\
&\overset{\text{Fact 2}}{\geq} \frac{L^2}{8} h^2(1-h)^2\left(1 - \mathrm{TV}\left(\mathbb{P}_g^{\otimes n}, \mathbb{P}_{f_{L,x_0,h}}^{\otimes n}\right)\right) \\
&\overset{\text{Pinsker}}{\geq} \frac{L^2}{8} h^2(1-h)^2\left(1 - \sqrt{\mathrm{KL}\left(\mathbb{P}_g^{\otimes n} \,\|\, \mathbb{P}_{f_{L,x_0,h}}^{\otimes n}\right)/2}\right) \\
&\overset{\text{Tensorization}}{=} \frac{L^2}{8} h^2(1-h)^2\left(1 - \sqrt{n\mathrm{KL}\left(\mathbb{P}_g \,\|\, \mathbb{P}_{f_{L,x_0,h}}\right)/2}\right) \\
&\overset{(17)}{\geq} \frac{L^2}{8} h^2(1-h)^2\left(1 - \sqrt{\frac{n}{2}\left(\frac{h^3 L^2}{3} + O(h^4)\right)}\right).
\end{aligned}
\tag{18}
$$

The second inequality comes from the so-called Le Cam's lemma Rigollet & Hütter (2015) that lower-bounds the testing difficulty (without further constraints) between two distributions. The next inequality comes from the so-called Pinsker's inequality Tsybakov (2009), that states that for two probability distributions $\mathbb{P}$ and $\mathbb{Q}$, $\mathrm{TV}\left(\mathbb{P}, \mathbb{Q}\right) \leq \sqrt{\mathrm{KL}\left(\mathbb{P} \,\|\, \mathbb{Q}\right)/2}$. The last inequality is the result of the so-called tensorization property of the KL divergence that states that for two probability distributions $\mathbb{P}$ and $\mathbb{Q}$, and for an integer $n \geq 1$, $\mathrm{KL}\left(\mathbb{P}^{\otimes n} \,\|\, \mathbb{Q}^{\otimes n}\right) \leq n\mathrm{KL}\left(\mathbb{P} \,\|\, \mathbb{Q}\right)$.

When possible (i.e. when $n$ is big enough), setting $h = \left(\frac{1}{4nL^2}\right)^{1/3}$ leads to, for $n$ big enough (so that $1 - h \geq 1/2$ and $|O(h^4)| \leq \frac{h^3 L^2}{3}$),

$$
\inf_{\hat{\pi} \text{ s.t. } \mathcal{C}} \sup_{\pi \in \Theta_L^{\mathrm{Lip}}} \mathbb{E}_{\mathbf{X} \sim \mathbb{P}_\pi^{\otimes n}, \hat{\pi}}\left(\left(\hat{\pi}(\mathbf{X})(x_0) - \pi(x_0)\right)^2\right) \geq \frac{L^2}{64}\left(\frac{1}{4L^2}\right)^{2/3} n^{-2/3}.
$$

This implies the first lower bound.

**$\epsilon$-DP overhead.** By Equation (18) and by Le Cam's lemma for differential privacy on product distributions (Fact 4),

$$
\inf_{\hat{\pi} \; \epsilon\text{-DP}} \sup_{\pi \in \Theta_L^{\mathrm{Lip}}} \mathbb{E}_{\mathbf{X} \sim \mathbb{P}_\pi^{\otimes n}, \hat{\pi}}\left(\left(\hat{\pi}(\mathbf{X})(x_0) - \pi(x_0)\right)^2\right) \geq \frac{L^2}{8} h^2(1-h)^2 e^{-n\epsilon \mathrm{TV}\left(\mathbb{P}_{f_{L,x_0,h}}, \mathbb{P}_g\right)}
$$

$$
\overset{(16)}{\geq} \frac{L^2}{8} h^2(1-h)^2 e^{-Ln\epsilon h^2}.
$$

When possible (i.e. when $n\epsilon$ is big enough), setting $h = 1/\sqrt{n\epsilon}$ leads to, when $n\epsilon$ is large enough to ensure $1 - h \geq 1/2$,

$$\inf_{\hat{\pi} \; \epsilon\text{-DP}} \sup_{\pi \in \Theta_L^{\text{Lip}}} \mathbb{E}_{\mathbf{X} \sim \mathbb{P}_\pi^{\otimes n}, \hat{\pi}} \left( (\hat{\pi}(\mathbf{X})(x_0) - \pi(x_0))^2 \right) \geq \frac{L^2 e^{-L}}{32} (n\epsilon)^{-1} .$$

**$\rho$-zCDP overhead.** By Le Cam's lemma for zero-concentrated differential privacy on product distributions (Fact 5) in (18),

$$\inf_{\hat{\pi} \; \epsilon\text{-DP}} \sup_{\pi \in \Theta_L^{\text{Lip}}} \mathbb{E}_{\mathbf{X} \sim \mathbb{P}_\pi^{\otimes n}, \hat{\pi}} \left( (\hat{\pi}(\mathbf{X})(x_0) - \pi(x_0))^2 \right) \geq \frac{L^2}{8} h^2 (1-h)^2 \left( 1 - n\sqrt{\rho/2} \text{TV} \left( \mathbb{P}_{f_{L,x_0,h}}, \mathbb{P}_g \right) \right)$$

$$\overset{(16)}{\geq} \frac{L^2}{8} h^2 (1-h)^2 \left( 1 - n\sqrt{\rho/2} L h^2 \right) .$$

When possible (i.e. when $n\sqrt{\rho}$ is large enough), setting $h = \left( \frac{1}{\sqrt{2}Ln\sqrt{\rho}} \right)^{1/2}$ leads to, when $n\sqrt{\rho}$ is large enough (so that $1 - h \geq 1/2$ ),

$$\inf_{\hat{\pi} \; \rho\text{-zCDP}} \sup_{\pi \in \Theta_L^{\text{Lip}}} \mathbb{E}_{\mathbf{X} \sim \mathbb{P}_\pi^{\otimes n}, \hat{\pi}} \left( (\hat{\pi}(\mathbf{X})(x_0) - \pi(x_0))^2 \right) \geq \frac{L}{64} (n\sqrt{\rho})^{-1} .$$

# F  Proof of Theorem 3

Let $m \in \mathbb{N} \setminus \{0\}$ that will be fixed later in the proof. As explained in the sketch of the proof, we build a packing consisting of functions that are parametrized by a vector $\omega \in \{0,1\}^m$. After controlling quantities such as their pairwise TV distances, and their KL divergences to the uniform distribution, we leverage Fano-type inequalities in order to obtain lower-bounds.

**Packing construction.** For any $\omega \in \{0,1\}^m$ different from 0 and any $h > 0$, we define the function $g_{L,\omega,h}$ as

$$g_{L,\omega,h} := \frac{1}{\|\omega\|_1} \sum_{i=1}^{m} \omega_i f_{\|\omega\|_1 L, \frac{i}{m+1}, h} , \tag{19}$$

where the functions $f$ are defined in (15). Note that $g_{L,\omega,h}$ is $L$-Lipschitz and that as soon as $h \leq h_m := \frac{1}{2(m+1)}$ it is also a valid density so that $g_{L,\omega,h} \in \Theta_L^{\text{Lip}}$. Notice that the function $g_{L,\omega,h}$ is constant to $1 - \|\omega\|_1 L h^2$ everywhere except on each interval $\left[ \frac{i}{m+1} - h, \frac{i}{m+1} + h \right]$ with $i$ such that $\omega_i \neq 0$, on which it deviates by a triangle of slopes $+L$ and $-L$.

By denoting by $K$ the triangle kernel such that $K(t) = Lt\mathbb{1}_{[-h,0]}(t) - Lt\mathbb{1}_{(0,h]}(t)$, it might be easier to visualize $g_{L,\omega,h}$ as

$$\forall t \in [0,1], \quad g_{L,\omega,h}(t) = 1 - \|\omega\|_1 \int K + \sum_{i=1}^{m} \omega_i K \left( t - \frac{1}{m+1} \right) , \tag{20}$$

where $\int K = L h^2$.

For $\omega, \omega' \in \{0,1\}^m$ and for $h$ small enough (i.e. $h \leq h_m$), we can bound the total variation between $\mathbb{P}_{g_{L,\omega,h}}$ and $\mathbb{P}_{g_{L,\omega',h}}$ as

$$
\begin{aligned}
\mathrm{TV}\left(\mathbb{P}_{g_{L,\omega,h}}, \mathbb{P}_{g_{L,\omega',h}}\right) &= \frac{1}{2}\int |g_{L,\omega,h} - g_{L,\omega',h}| \\
&\overset{(20)}{=} \frac{1}{2}\int \left| (\|\omega'\|_1 - \|\omega\|_1)\int K + \sum_{i=1}^{m}(\omega'_i - \omega_i)K\left(\cdot - \frac{1}{m+1}\right)\right| \\
&\leq \frac{1}{2}\int \left| \|\omega'\|_1 - \|\omega\|_1 \right| \int K + \sum_{i=1}^{m}|\omega'_i - \omega_i|K\left(\cdot - \frac{1}{m+1}\right) \\
&= \frac{1}{2}\left( \left| \|\omega'\|_1 - \|\omega\|_1 \right| + d_{\mathrm{ham}}(\omega, \omega') \right)\int K && (21) \\
&\leq mLh^2 . && (22)
\end{aligned}
$$

The KL divergence between $\mathbb{P}_{g_{L,\omega,h}}$ and $\mathbb{P}_g$, with $g$ the density constant equal to 1 on $[0,1]$, satisfies

$$
\begin{aligned}
\mathrm{KL}\left(\mathbb{P}_{g_{L,\omega,h}} \,\|\, \mathbb{P}_g\right) &= \int_{[0,1]} \ln\left(g_{L,\omega,h}\right) g_{L,\omega,h} \\
&= \ln\left(1 - \|\omega\|_1 Lh^2\right)\left(1 - \|\omega\|_1 Lh^2\right)\left(1 - \|\omega\|_1 2h\right) \\
&\quad + 2\|\omega\|_1 \int_0^h \ln\left(1 - \|\omega\|_1 Lh^2 + Lt\right)\left(1 - \|\omega\|_1 Lh^2 + Lt\right) dt \\
&\overset{\ln(1+\cdot)\leq\cdot}{\leq} \left(-\|\omega\|_1 Lh^2\right)\left(1 - \|\omega\|_1 Lh^2\right)\left(1 - \|\omega\|_1 2h\right) \\
&\quad + 2\|\omega\|_1 \int_0^h \left(-\|\omega\|_1 Lh^2 + Lt\right)\left(1 - \|\omega\|_1 Lh^2 + Lt\right) dt \\
&\overset{\text{Calculus}}{=} \frac{L^2}{3}\|\omega\|_1 h^3(2 - 3\|\omega\|_1 h) .
\end{aligned}
\quad (23)
$$

Finally, we lower bound the squared $L^2$ distance between $g_{L,\omega,h}$ and $g_{L,\omega',h}$:

$$
\begin{aligned}
\int_{[0,1]} & (g_{L,\omega,h} - g_{L,\omega',h})^2 \\
&= \sum_{i=1}^{m} \mathbb{1}_{\omega_i \neq \omega'_i} \int_{\frac{i}{m+1}-h}^{\frac{i}{m+1}+h} \left( (\|\omega'\|_1 - \|\omega\|_1)\int K + (\omega_i - \omega'_i)K\left(t - \frac{i}{m+1}\right)\right)^2 dt \\
&\geq \sum_{i=1}^{m} \mathbb{1}_{\omega_i \neq \omega'_i} \int_{\frac{i}{m+1}-h}^{\frac{i}{m+1}+h} \left( K\left(t - \frac{i}{m+1}\right) - \left| \|\omega\|_1 - \|\omega'\|_1 \right| \int K\right)^2 dt \\
&\geq \sum_{i=1}^{m} \mathbb{1}_{\omega_i \neq \omega'_i} \int_{\frac{i}{m+1}-h}^{\frac{i}{m+1}+h} \left\{ \left( K\left(t - \frac{i}{m+1}\right)\right)^2 \right. \\
&\qquad\qquad \left. -2K\left(t - \frac{i}{m+1}\right)\left| \|\omega\|_1 - \|\omega'\|_1 \right| \int K \right\} dt \\
&\geq d_{\mathrm{ham}}(\omega, \omega')\left(\int K^2 - 2m\left(\int K\right)^2\right) \\
&\geq 2d_{\mathrm{ham}}(\omega, \omega') L^2 \left(\frac{h^3}{3} - mh^4\right) \\
&= \frac{2d_{\mathrm{ham}}(\omega, \omega') L^2 h^3 (1 - 3mh)}{3} .
\end{aligned}
\quad (24)
$$

By the Varshamov-Gilbert theorem (Tsybakov, 2009, Lemma 2.7), as long as $m \geq 8$, there exist $M \in \mathbb{N}$ and $\omega^{(0)}, \ldots, \omega^{(M)} \in \{0,1\}^m$ such that $M \geq 2^{m/8}$, $\omega^{(0)} = \{0\}^m$ and $i \neq j \implies d_{\text{ham}}\left(\omega^{(i)}, \omega^{(j)}\right) \geq m/8$. According to (24), the family $\left(g_{L,\omega^{(i)},h}\right)_{i=1,\ldots,M}$ is then an $\Omega := \frac{1}{2}\sqrt{\frac{mL^2(h^3 - 3mh^4)}{12}}$ packing of $\Theta_L^{\text{Lip}}$ for the $L^2$ distance.

**Recovering the usual lower-bound.** By Equation (5) with $\Phi(\cdot) := (\cdot)^2$ and $\|\cdot\|$ the $L^2$ norm,

$$
\begin{aligned}
\inf_{\hat{\pi} \text{ s.t. } \mathcal{C}} \sup_{\pi \in \Theta_L^{\text{Lip}}} &\mathbb{E}_{\mathbf{X} \sim \mathbb{P}_\pi^{\otimes n}, \hat{\pi}} \left( \int_{[0,1]} (\hat{\pi}(\mathbf{X}) - \pi)^2 \right) \\
&\geq \frac{mL^2 h^3 (1 - 3mh)}{48} \inf_{\hat{\pi} \text{ s.t. } \mathcal{C}} \inf_{\Psi : \Theta_L^{\text{Lip}} \to \{0,1\}} \max_{i=1,\ldots,M} \mathbb{P}_{\mathbf{X} \sim \mathbb{P}_{g_{L,\omega^{(i)},h}}^{\otimes n}, \hat{\pi}} \left( \Psi(\hat{\pi}(\mathbf{X})) \neq i \right) \\
&\overset{\text{Fact 3}}{\geq} \frac{mL^2 h^3 (1 - 3mh)}{48} \left( 1 - \frac{1 + \frac{1}{M}\sum_{1 \leq i \leq M} \text{KL}\left( \mathbb{P}_{g_{L,\omega^{(i)},h}}^{\otimes n} \middle\| \mathbb{P}_g^{\otimes n} \right)}{\ln(M)} \right) \\
&\overset{\text{Tensorization}}{=} \frac{mL^2 h^3 (1 - 3mh)}{48} \left( 1 - \frac{1 + \frac{n}{M}\sum_{1 \leq i \leq M} \text{KL}\left( \mathbb{P}_{g_{L,\omega^{(i)},h}} \middle\| \mathbb{P}_g \right)}{\ln(M)} \right) \\
&\overset{(23)\&\|\omega\|_1 \leq m, M \geq 2^{m/8}}{\geq} \frac{mL^2 h^3 (1 - 3mh)}{48} \left( 1 - \frac{1 + \frac{L^2}{3}nmh^3(2 - 3mh)}{\ln(2)m/8} \right) .
\end{aligned}
\tag{25}
$$

So, by choosing $m = \lceil n^{1/3} \rceil$ and $h = \frac{c}{m}$ where c is a positive constant small enough we get, for $n$ big enough,

$$
\inf_{\hat{\pi} \text{ } \epsilon\text{-DP}} \sup_{\pi \in \Theta_L^{\text{Lip}}} \mathbb{E}_{\mathbf{X} \sim \mathbb{P}_\pi^{\otimes n}, \hat{\pi}} \left( \int_{[0,1]} (\hat{\pi}(\mathbf{X}) - \pi)^2 \right) \geq C^{-1}(n)^{-2/3} ,
$$

where $C$ is a positive constant depending only on $L$.

$\epsilon$**-DP overhead.** By the same reduction and Fano's lemma for differential privacy on product distributions (Fact 6), we get for any $h \leq h_m$,

$$
\begin{aligned}
\inf_{\hat{\pi} \text{ } \epsilon\text{-DP}} \sup_{\pi \in \Theta_L^{\text{Lip}}} &\mathbb{E}_{\mathbf{X} \sim \mathbb{P}_\pi^{\otimes n}, \hat{\pi}} \left( \int_{[0,1]} (\hat{\pi}(\mathbf{X}) - \pi)^2 \right) \\
&\geq \frac{mL^2 h^3 (1 - 3mh)}{48} \left( 1 - \frac{1 + \frac{n\epsilon}{M^2} 2\sum_{1 \leq i,j \leq M} \text{TV}\left( \mathbb{P}_{g_{L,\omega^{(i)},h}}, \mathbb{P}_{g_{L,\omega^{(j)},h}} \right)}{\ln(M)} \right) \\
&\overset{(22)\& M \geq 2^{m/8}}{\geq} \frac{mL^2 h^3 (1 - 3mh)}{48} \left( 1 - \frac{1 + 2n\epsilon mLh^2}{\ln(2)m/8} \right) .
\end{aligned}
$$

So, by choosing $m = \lceil \sqrt{n\epsilon} \rceil$ and $h = \frac{c}{m}$ where c is small enough a positive constant (depending only on $L$), we get, as soon as $\min(n, n\epsilon)$ is big enough,

$$
\inf_{\hat{\pi} \text{ } \epsilon\text{-DP}} \sup_{\pi \in \Theta_L^{\text{Lip}}} \mathbb{E}_{\mathbf{X} \sim \mathbb{P}_\pi^{\otimes n}, \hat{\pi}} \left( \int_{[0,1]} (\hat{\pi}(\mathbf{X}) - \pi)^2 \right) \geq C'^{-1}(n\epsilon)^{-1} ,
$$

where $C'$ is a positive constant depending only on $L$.

$\rho$**-zCDP overhead.** For $\rho$-zCDP, we present the proof using both Fano's lemma and Assouad's method. We will see that Assouad gives better results

**Fano version.** By again the same reduction and Fano's lemma for zero-concentrated differential privacy (Fact 7), denoting $t_{i,j} := \text{TV}\left(\mathbb{P}_{g_{L,\omega^{(i)},h}}, \mathbb{P}_{g_{L,\omega^{(j)},h}}\right)$, we get for any $h \leq h_m$,

$$\inf_{\hat{\pi} \ \rho\text{-zCDP}} \sup_{\pi \in \Theta_L^{\text{Lip}}} \mathbb{E}_{\mathbf{X} \sim \mathbb{P}_\pi^{\otimes n}, \hat{\pi}} \left( \int_{[0,1]} (\hat{\pi}(\mathbf{X}) - \pi)^2 \right)$$

$$\geq \frac{mL^2 h^3 (1 - 3mh)}{48} \left( 1 - \frac{1 + \frac{n^2 \rho}{M^2} 4 \sum_{1 \leq i,j \leq M} \left( \frac{1}{n} t_{i,j} + t_{i,j}^2 \right)}{\ln(M)} \right)$$

$$\overset{(22)}{\geq} \frac{mL^2 h^3 (1 - 3mh)}{48} \left( 1 - \frac{1 + n^2 \rho 4 \left( \frac{mLh^2}{n} + m^2 L^2 h^4 \right)}{\ln(2) m/8} \right) .$$

So, by choosing $m = \left\lceil \left( n\sqrt{\rho} \right)^{\frac{2}{3}} \right\rceil$ and $h = \frac{c}{m}$ for $c$ small enough (depending only on $L$), if $\frac{n}{\rho}$ is big enough, we get that

$$\inf_{\hat{\pi} \ \text{s.t.} \ \rho\text{-zCDP}} \sup_{\pi \in \Theta_L^{\text{Lip}}} \mathbb{E}_{\mathbf{X} \sim \mathbb{P}_\pi^{\otimes n}, \hat{\pi}} \left( \int_{[0,1]} (\hat{\pi}(\mathbf{X}) - \pi)^2 \right) \geq C''^{-1} (n\sqrt{\rho})^{-4/3}$$

where $C''$ is a positive constant depending only on $L$.

**Assouad version.** From Equation (24), we can see that when $h := \frac{c}{m}$ for a positive $c$ that is small enough, the condition expressed in Equation (12) is satisfied for $\tau = \Omega(h^3)$. To apply (14), the only missing ingredient is to bound the testing difficulties between the mixtures on the hypercube.

In the sequel, $\mathbb{P}_\omega$ is used as a short for $\mathbb{P}_{g_{L,\omega,h}}$. We need to bound the total variation between the mixtures on the hypercube (see (14)) as

$$\text{TV}\left(\mathbb{P}_{\omega^{i,0}}, \mathbb{P}_{\omega^{i,1}}\right) = \text{TV}\left( \frac{1}{2^{m-1}} \sum_{\omega \in \{0,1\}^m | \omega_i = 0} \mathbb{P}_{g_{L,\omega,h}}, \frac{1}{2^{m-1}} \sum_{\omega \in \{0,1\}^m | \omega_i = 1} \mathbb{P}_{g_{L,\omega,h}} \right)$$

$$= \frac{1}{2} \frac{1}{2^{m-1}} \int \left| \sum_{\omega \in \{0,1\}^m | \omega_i = 0} g_{L,\omega,h} - \sum_{\omega \in \{0,1\}^m | \omega_i = 1} g_{L,\omega,h} \right|$$

$$= \frac{1}{2^m} \int \left| \sum_{\omega_1,\ldots,\omega_{i-1},\omega_{i+1}\ldots,\omega_m \in \{0,1\}} \left( g_{L,(\omega_1,\ldots,\omega_{i-1},0,\omega_{i+1}\ldots,\omega_m),h} - \right. \right.$$

$$\left. \left. g_{L,(\omega_1,\ldots,\omega_{i-1},1,\omega_{i+1}\ldots,\omega_m),h} \right) \right|$$

$$\leq \frac{1}{2^m} \sum_{\omega_1,\ldots,\omega_{i-1},\omega_{i+1}\ldots,\omega_m \in \{0,1\}} \int \left| g_{L,(\omega_1,\ldots,\omega_{i-1},0,\omega_{i+1}\ldots,\omega_m),h} - \right.$$

$$\left. g_{L,(\omega_1,\ldots,\omega_{i-1},1,\omega_{i+1}\ldots,\omega_m),h} \right|$$

$$\overset{(21)}{\leq} \frac{1}{2^m} \sum_{\omega_1,\ldots,\omega_{i-1},\omega_{i+1}\ldots,\omega_m \in \{0,1\}} 2Lh^2$$

$$= O\left(h^2\right) .$$

Here and in the sequel, the asymptotic comparators only hide constants and terms that depend on $L$. All in all, by using Le Cam's lemma for product distribution and $\rho$-zCDP Fact 5, and by Equation (13), since $\tau = \Omega(h^3)$ we obtain

$$\inf_{\hat{\pi} \ \rho\text{-zCDP}} \sup_{\pi \in \Theta_{L,\beta}^{\text{PSob}}} \mathbb{E}_{\mathbf{X} \sim \mathbb{P}_\pi^{\otimes n}, \hat{\pi}} \left( \int_{[0,1]} (\hat{\pi}(\mathbf{X}) - \pi)^2 \right) = \Omega\left(mh^3\right) \left( 1 - n\sqrt{\rho} O\left(h^2\right) \right) . \tag{26}$$

Setting $h \approx \left( n\sqrt{\rho} \right)^{\frac{-1}{2}}$ concludes the proof.

## G   proof of Lemma 2

Let $\pi \in \Theta^{\mathrm{PSob}}_{L,\beta}$. We have,

$$\mathbb{E}\left( \int_{[0,1]} \left( \hat{\pi}^{\mathrm{proj}}(\mathbf{X}) - \pi \right)^2 \right) \overset{\text{Parseval}}{=} \mathbb{E}\left( \sum_{i=1}^{N} \left( \hat{\theta}_i - \theta_i + \frac{1}{n}Z_i \right)^2 + \sum_{i=N+1}^{+\infty} \theta_i^2 \right)$$

$$= \sum_{i=1}^{N} \mathbb{E}\left( \left( \hat{\theta}_i - \theta_i + \frac{1}{n}Z_i \right)^2 \right) + \sum_{i=N+1}^{+\infty} \theta_i^2 \ .$$

Furthermore, for any $i$, since $Z$ is centered

$$\mathbb{E}\left( \hat{\theta}_i \right) = \mathbb{E}\left( \frac{1}{n} \sum_{j=1}^{n} \phi_i(X_j) \right) = \frac{1}{n} \sum_{j=1}^{n} \mathbb{E}\left( \phi_i(X_j) \right) \overset{X_j \text{ i.i.d.}}{=} \mathbb{E}_{\mathbf{X} \sim \mathbb{P}_\pi^{\otimes n}} \phi_i(X_1) = \int \pi\phi_i = \theta_i$$

Hence, for any $i$, since $Z_i$ is independent from the dataset

$$\mathbb{E}\left( \left( \hat{\theta}_i - \theta_i + \frac{1}{n}Z_i \right)^2 \right) = \mathbb{V}\left( \hat{\theta}_i \right) + \frac{1}{n^2}\mathbb{V}\left( Z_i \right) \overset{\text{Independence of } X_j}{=} \frac{1}{n^2} \sum_{j=1}^{n} \mathbb{V}\left( \phi_i\left( X_j \right) \right) + \frac{1}{n^2}\mathbb{V}\left( Z_i \right)$$

$$\overset{|\phi_i| \leq \sqrt{2}}{\leq} \frac{2}{n} + \frac{1}{n^2}\mathbb{V}\left( Z \right) \ .$$

Finally, with $a_j := j - 1$, Fact 1 allows bounding $\sum_{i=m+1}^{+\infty} \theta_i^2$ as

$$\sum_{i=N+1}^{+\infty} \theta_i^2 \leq \frac{1}{N^{2\beta}} \sum_{i=N+1}^{+\infty} a_i^{2\beta} \theta_i^2 \leq \frac{1}{N^{2\beta}} \sum_{i=1}^{+\infty} a_i^{2\beta} \theta_i^2 \overset{Fact\ 1}{\leq} \frac{1}{N^{2\beta}} \frac{L^2}{\pi^{2\beta}} \ .$$

This yields the conclusion with $C_{L,\beta} := \max(2, L^2/\pi^{2\beta})$.

## H   Proof of Theorem 5

Let us consider the following well-known function :

$$\forall x \in \mathbb{R}, \quad K_0(x) := e^{-\frac{1}{1-x^2}} \mathbb{1}_{(-1,1)}(x) \ .$$

We can notice that for any $\beta > 0$ there exists $\nu > 0$ such that the kernel $K(x) := \nu K_0(2x)$ satisfies $K \in \mathcal{C}^\infty(\mathbb{R}, [0, +\infty))$, $\int \left( K^{(\beta)} \right)^2 \leq 1$ and $K(x) > 0$ iff $x \in (-1/2, 1/2)$. Furthermore, for any $i \in \mathbb{N}$, $K^{(i)}(x) = 0$ for every $x \in (-\infty, -1/2] \cup [1/2, +\infty)$.

**Packing construction.** Let $m \in \mathbb{N} \setminus \{0\}$ that will be fixed later. For any $h > 0$, and $\omega \in \{0,1\}^m$, we define the function $g_{L,\beta,\omega,h}$ as,

$$\forall x \in [0,1], \quad g_{L,\beta,\omega,h}(x) := 1 - \|\omega\|_1 Lh^{\beta+1} \int K + Lh^\beta \sum_{i=1}^{m} \omega_i K\left( \frac{x - \frac{i}{m+1}}{h} \right) \ . \tag{27}$$

Note that when $h < \frac{1}{m+1}$ we have $\int_0^1 g_{L,\beta,\omega,h} = 1$; when $mh \int \left( K^{(\beta)} \right)^2 \leq 1$, we have $g_{L,\beta,\omega,h} \geq 0$; and when both hold we have $g_{L,\beta,\omega,h} \in \Theta^{\mathrm{PSob}}_{L,\beta}$ (see Equation (9)). Indeed, under these hypotheses, the periodicity conditions are immediate (the function is constant on neighborhoods of 0 and 1, with the same value). The

energy of the $\beta$th derivative can be bounded as

$$
\begin{aligned}
\int \left( g_{L,\beta,\omega,h}^{(\beta)} \right)^2 &= \int \left( Lh^\beta \sum_{i=1}^m \omega_i \left( x \mapsto K \left( \frac{x - \frac{i}{m+1}}{h} \right) \right)^{(\beta)} \right)^2 \\
&= \int \left( L \sum_{i=1}^m \omega_i K^{(\beta)} \left( \frac{\cdot - \frac{i}{m+1}}{h} \right) \right)^2 \\
&\stackrel{\text{disjoint support}}{=} L^2 \sum_{i=1}^m \omega_i \int \left( K^{(\beta)} \left( \frac{\cdot - \frac{i}{m+1}}{h} \right) \right)^2 \\
&= L^2 m h \int \left( K^{(\beta)} \right)^2 \leq L^2 \, .
\end{aligned}
$$

In the sequel of this proof, this hypothesis will always be satisfied asymptotically for all the values of $m$ and $h$ that will be considered. From now on, we may consider it valid.

Given $h > 0$ and $\omega, \omega' \in \{0, 1\}^m$, when $g_{L,\beta,\omega,h}, g_{L,\beta,\omega',h} \in \Theta_{L,\beta}^{\mathrm{PSob}}$, we can bound the total variation between $\mathbb{P}_{g_{L,\beta,\omega,h}}$ and $\mathbb{P}_{g_{L,\beta,\omega',h}}$ as,

$$
\begin{aligned}
\mathrm{TV} \left( \mathbb{P}_{g_{L,\beta,\omega,h}}, \mathbb{P}_{g_{L,\beta,\omega',h}} \right) &= \frac{1}{2} \int \left| g_{L,\beta,\omega,h} - g_{L,\beta,\omega',h} \right| \\
&= \frac{1}{2} \int \left| (\|\omega'\|_1 - \|\omega\|_1) L h^{\beta+1} \int K + \sum_{i=1}^m (\omega_i' - \omega_i) L h^\beta K \left( \frac{\cdot - \frac{1}{m+1}}{h} \right) \right| \\
&\leq \frac{1}{2} \int \left| \|\omega'\|_1 - \|\omega\|_1 \right| L h^{\beta+1} \int K + \sum_{i=1}^m |\omega_i' - \omega_i| L h^\beta K \left( \frac{\cdot - \frac{1}{m+1}}{h} \right) \\
&= \frac{1}{2} \left( \left| \|\omega'\|_1 - \|\omega\|_1 \right| + d_{\mathrm{ham}} (\omega, \omega') \right) L h^{\beta+1} \int K \qquad (28) \\
&\leq m L h^{\beta+1} \, . \qquad (29)
\end{aligned}
$$

The KL divergence between $\mathbb{P}_{g_{L,\beta,\omega,h}}$ and $\mathbb{P}_g$, the uniform distribution on $[0,1]$, is bounded as

$$
\begin{aligned}
\mathrm{KL}\left(\mathbb{P}_{g_{L,\beta,\omega,h}} \,\|\, \mathbb{P}_g\right) &= \int_{[0,1]} \ln\left(g_{L,\beta,\omega,h}\right) g_{L,\beta,\omega,h} \\
&= \int_{[0,1]\setminus\cup_{i:\omega_i\neq 0}\left[\frac{i}{m+1}-\frac{h}{2},\frac{i}{m+1}+\frac{h}{2}\right]} \ln\left(1 - \|\omega\|_1 L h^{\beta+1}\int K\right)\left(1 - \|\omega\|_1 L h^{\beta+1}\int K\right)dt \\
&\quad + \|\omega\|_1 \int_{-\frac{h}{2}}^{\frac{h}{2}} \ln\left(1 - \|\omega\|_1 L h^{\beta+1}\int K + L h^\beta K\left(\frac{t}{h}\right)\right) \\
&\qquad\qquad \left(1 - \|\omega\|_1 L h^{\beta+1}\int K + L h^\beta K\left(\frac{t}{h}\right)\right)dt \\
&\overset{\ln(1+\cdot)\leq\cdot}{\leq} (1 - \|\omega\|_1 h)\left(-\|\omega\|_1 L h^{\beta+1}\int K\right)\left(1 - \|\omega\|_1 L h^{\beta+1}\int K\right) \\
&\quad + \|\omega\|_1 \int_{-\frac{h}{2}}^{\frac{h}{2}}\left(-\|\omega\|_1 L h^{\beta+1}\int K + L h^\beta K\left(\frac{t}{h}\right)\right) \\
&\qquad\qquad \left(1 - \|\omega\|_1 L h^{\beta+1}\int K + L h^\beta K\left(\frac{t}{h}\right)\right)dt \\
&\overset{\text{Calculus}}{=} \|\omega\|_1 L^2 h^{2\beta+1}\int K^2 - \|\omega\|_1^2 L^2 h^{2\beta+2}\int K \\
&\leq \|\omega\|_1 L^2 h^{2\beta+1}\int K^2 \leq m L^2 h^{2\beta+1}\int K^2 \,.
\end{aligned}
\tag{30}
$$

Finally, the squared $L^2$ distance between $g_{L,\beta,\omega,h}$ and $g_{L,\beta,\omega',h}$ can be lower bounded as,

$$
\begin{aligned}
\int_{[0,1]} &\left(g_{L,\beta,\omega,h} - g_{L,\beta,\omega',h}\right)^2 \\
&= \sum_{i=1}^m \mathbb{1}_{\omega_i\neq\omega_i'} \int_{\frac{i}{m+1}-\frac{h}{2}}^{\frac{i}{m+1}+\frac{h}{2}}\left(L h^{\beta+1}(\|\omega'\|_1 - \|\omega\|_1)\int K + (\omega_i - \omega_i') L h^\beta K\left(\frac{t-\frac{i}{m+1}}{h}\right)\right)^2 dt \\
&\geq \sum_{i=1}^m \mathbb{1}_{\omega_i\neq\omega_i'} \int_{\frac{i}{m+1}-\frac{h}{2}}^{\frac{i}{m+1}+\frac{h}{2}}\left(L h^\beta K\left(\frac{t-\frac{i}{m+1}}{h}\right) - L h^{\beta+1}\left|\|\omega\|_1 - \|\omega'\|_1\right|\int K\right)^2 dt \\
&\geq \sum_{i=1}^m \mathbb{1}_{\omega_i\neq\omega_i'} \int_{\frac{i}{m+1}-\frac{h}{2}}^{\frac{i}{m+1}+\frac{h}{2}}\left\{\left(L h^\beta K\left(\frac{t-\frac{i}{m+1}}{h}\right)\right)^2 \right. \\
&\qquad\qquad \left. -2 L h^\beta K\left(\frac{t-\frac{i}{m+1}}{h}\right) L h^{\beta+1}\left|\|\omega\|_1 - \|\omega'\|_1\right|\int K\right\} dt \\
&\geq d_{\mathrm{ham}}\left(\omega,\omega'\right) L^2 h^{2\beta+1}\left(\int K^2 - 2mh\left(\int K\right)^2\right)\,.
\end{aligned}
\tag{31}
$$

By the Varshamov-Gilbert theorem (Tsybakov, 2009, Lemma 2.7), as long as $m \geq 8$, there exist $M \in \mathbb{N}$ and $\omega^{(0)},\ldots,\omega^{(M)} \in \{0,1\}^m$ such that $M \geq 2^{m/8}$, $\omega^{(0)} = \{0\}^m$ and $i \neq j \implies d_{\mathrm{ham}}\left(\omega^{(i)},\omega^{(j)}\right) \geq m/8$. According to (31), the family $\left(g_{L,\beta,\omega^{(i)},h}\right)_{i=1,\ldots,M}$ is then a $\Omega = \frac{1}{2}\sqrt{\frac{m}{8} L^2 h^{2\beta+1}\left(\int K^2 - 2mh\left(\int K\right)^2\right)}$ packing of $\Theta_{L,\beta}^{\mathrm{PSob}}$ for the $L^2$ distance.

**Recovering the usual lower-bound.** By Equation (5) with $\Phi(\cdot) := (\cdot)^2$ and $\|\cdot\|$ the $L^2$ norm,

$$\inf_{\hat{\pi} \text{ s.t. } \mathcal{C}} \sup_{\pi \in \Theta^{\text{PSob}}_{L,\beta}} \mathbb{E}_{\mathbf{X} \sim \mathbb{P}^{\otimes n}_\pi, \hat{\pi}} \int_{[0,1]} (\hat{\pi}(\mathbf{X}) - \pi)^2$$

$$\geq \frac{L^2}{32} m h^{2\beta+1} \left( \int K^2 - 2mh \left( \int K \right)^2 \right)$$

$$\inf_{\hat{\pi} \text{ s.t. } \mathcal{C}} \inf_{\Psi:\Theta^{\text{PSob}}_{L,\beta} \to \{0,1\}} \max_{i=1,\dots,M} \mathbb{P}_{\mathbf{X} \sim \mathbb{P}^{\otimes n}_{g_{L,\beta,\omega(i)},h}, \hat{\pi}} \left( \Psi(\hat{\pi}(\mathbf{X})) \neq i \right)$$

$$\stackrel{\text{Fact 3}}{\geq} \frac{L^2}{32} m h^{2\beta+1} \left( \int K^2 - 2mh \left( \int K \right)^2 \right)$$

$$\left( 1 - \frac{1 + \frac{1}{M} \sum_{1 \leq i \leq M} \text{KL} \left( \mathbb{P}^{\otimes n}_{g_{L,\beta,\omega(i)},h} \middle\| \mathbb{P}^{\otimes n}_g \right)}{\ln(M)} \right) \tag{32}$$

$$\stackrel{\text{Tensorization}}{=} \frac{L^2}{32} m h^{2\beta+1} \left( \int K^2 - 2mh \left( \int K \right)^2 \right)$$

$$\left( 1 - \frac{1 + \frac{n}{M} \sum_{1 \leq i \leq M} \text{KL} \left( \mathbb{P}_{g_{L,\beta,\omega(i)},h} \middle\| \mathbb{P}_g \right)}{\ln(M)} \right)$$

$$\stackrel{(30)\&M \geq 2^{m/8}}{\geq} \frac{L^2}{32} m h^{2\beta+1} \left( \int K^2 - 2mh \left( \int K \right)^2 \right) \left( 1 - \frac{1 + nmL^2 h^{2\beta+1} \int K^2}{\ln(2)m/8} \right) .$$

Finally, setting $m = \left\lceil n^{\frac{1}{2\beta+1}} \right\rceil$ and $h = \frac{c}{m}$ for $c$ small enough gives that, for $n$ big enough,

$$\inf_{\hat{\pi} \text{ } \epsilon\text{-DP}} \sup_{\pi \in \Theta^{\text{PSob}}_{L,\beta}} \mathbb{E}_{\mathbf{X} \sim \mathbb{P}^{\otimes n}_\pi, \hat{\pi}} \int_{[0,1]} (\hat{\pi}(\mathbf{X}) - \pi)^2 \geq C^{-1} n^{-\frac{2\beta}{2\beta+1}} ,$$

where $C$ is a positive constant depending only on $L$ and $\beta$.

**$\epsilon$-DP overhead.** By the same reduction and Fano's lemma for differential privacy on product distributions (Fact 6), we get

$$\inf_{\hat{\pi} \text{ } \epsilon\text{-DP}} \sup_{\pi \in \Theta^{\text{PSob}}_{L,\beta}} \mathbb{E}_{\mathbf{X} \sim \mathbb{P}^{\otimes n}_\pi, \hat{\pi}} \int_{[0,1]} (\hat{\pi}(\mathbf{X}) - \pi)^2$$

$$\geq \frac{L^2}{32} m h^{2\beta+1} \left( \int K^2 - 2mh \left( \int K \right)^2 \right)$$

$$\left( 1 - \frac{1 + \frac{n\epsilon}{M^2} 2 \sum_{1 \leq i,j \leq M} \text{TV} \left( \mathbb{P}_{g_{L,\beta,\omega(i)},h}, \mathbb{P}_{g_{L,\beta,\omega(j)},h} \right)}{\ln(M)} \right)$$

$$\stackrel{(29)}{\geq} \frac{L^2}{32} m h^{2\beta+1} \left( \int K^2 - 2mh \left( \int K \right)^2 \right) \left( 1 - \frac{1 + 2n\epsilon mL h^{\beta+1} \int K}{\ln(2)m/8} \right) .$$

Setting $m = \left\lceil (n\epsilon)^{\frac{1}{\beta+1}} \right\rceil$ and $h = \frac{c}{m}$ for $c$ small enough leads to, for $n\epsilon$ big enough,

$$\inf_{\hat{\pi} \text{ } \epsilon\text{-DP}} \sup_{\pi \in \Theta^{\text{PSob}}_{L,\beta}} \mathbb{E}_{\mathbf{X} \sim \mathbb{P}^{\otimes n}_\pi, \hat{\pi}} \int_{[0,1]} (\hat{\pi}(\mathbf{X}) - \pi)^2 \geq C'^{-1} (n\epsilon)^{-\frac{2\beta}{\beta+1}} ,$$

where $C'$ is a constant depending only on $L$ and $\beta$.

$\rho$**-zCDP overhead.** For $\rho$-zCDP, we present the proof using both Fano's lemma and Assouad's method. We will see that Assouad gives better results.

**Fano version.** By again the same reduction and Fano's lemma for zero-concentrated differential privacy (Fact 7), denoting $t_{i,j} := \mathrm{TV}\left(\mathbb{P}_{g_{L,\beta,\omega^{(i)},h}}, \mathbb{P}_{g_{L,\beta,\omega^{(j)},h}}\right)$, we get

$$
\inf_{\hat{\pi} \ \rho\text{-zCDP}} \sup_{\pi \in \Theta_{L,\beta}^{\mathrm{PSob}}} \mathbb{E}_{\mathbf{X} \sim \mathbb{P}_\pi^{\otimes n}, \hat{\pi}} \int_{[0,1]} (\hat{\pi}(\mathbf{X}) - \pi)^2
$$

$$
\geq \frac{L^2}{32} m h^{2\beta+1} \left( \int K^2 - 2mh \left( \int K \right)^2 \right)
$$

$$
\left( 1 - \frac{1 + \frac{n^2 \rho}{M^2} 4 \sum_{1 \leq i,j \leq M} \frac{1}{n} t_{i,j} + t_{i,j}^2}{\ln(M)} \right)
$$

$$
\overset{(29)}{\geq} \frac{L^2}{32} m h^{2\beta+1} \left( \int K^2 - 2mh \left( \int K \right)^2 \right)
$$

$$
\left( 1 - \frac{1 + 4n^2 \rho \left( \frac{mLh^{\beta+1} \int K}{n} + \left( mLh^{\beta+1} \int K \right)^2 \right)}{\ln(2) m / 8} \right).
$$

So, by choosing $m = \left\lceil \left( n\sqrt{\rho} \right)^{\frac{2}{2\beta+1}} \right\rceil$ and $h = \frac{c}{m}$ for $c$ small enough, if $n\sqrt{\rho}$ and $\frac{n}{\left( n\sqrt{\rho} \right)^{\frac{2\beta}{2\beta+1}}} = \left( n\sqrt{\rho} \right)^{\frac{1}{2\beta+1}} / \sqrt{\rho}$ are big enough,

$$
\inf_{\hat{\pi} \ \epsilon\text{-DP}} \sup_{\pi \in \Theta_{L,\beta}^{\mathrm{PSob}}} \mathbb{E}_{\mathbf{X} \sim \mathbb{P}_\pi^{\otimes n}, \hat{\pi}} \int_{[0,1]} (\hat{\pi}(\mathbf{X}) - \pi)^2 \geq C''^{-1} \left( n\sqrt{\rho} \right)^{-\frac{2\beta}{\beta+1/2}},
$$

where $C''$ is a constant depending only on $L$ and $\beta$.

**Assouad version.** From Equation (31), we can see that when $h := \frac{c}{m}$ for a positive $c$ that is small enough, the condition expressed in Equation (12) is satisfied for $\tau = \Omega(h^{2\beta+1})$. To apply (14), the only missing ingredient is to bound the testing difficulties between the mixtures on the hypercube.

In the sequel, $\mathbb{P}_\omega$ is used as a short for $\mathbb{P}_{g_{L,\beta,\omega,h}}$. We need to bound the total variation between the mixtures on the hypercube (denoted $\mathbb{P}_{\omega^{i,0}}$ and $\mathbb{P}_{\omega^{i,1}}$, cf (14)) as

$$
\begin{aligned}
\mathrm{TV}&\left(\mathbb{P}_{\omega^{i,0}}, \mathbb{P}_{\omega^{i,1}}\right) \\
&= \frac{1}{2}\frac{1}{2^{m-1}} \int \left| \sum_{\omega \in \{0,1\}^m | \omega_i = 0} g_{L,\beta,\omega,h} - \sum_{\omega \in \{0,1\}^m | \omega_i = 1} g_{L,\beta,\omega,h} \right| \\
&= \frac{1}{2^m} \int \left| \sum_{\omega_1,\ldots,\omega_{i-1},\omega_{i+1}\ldots,\omega_m \in \{0,1\}} \left( g_{L,\beta,(\omega_1,\ldots,\omega_{i-1},0,\omega_{i+1}\ldots,\omega_m),h} - \right. \right. \\
&\qquad\qquad\qquad\qquad\qquad\qquad\qquad\qquad\qquad \left.\left. g_{L,\beta,(\omega_1,\ldots,\omega_{i-1},1,\omega_{i+1}\ldots,\omega_m),h} \right) \right| \\
&\leq \frac{1}{2^m} \sum_{\omega_1,\ldots,\omega_{i-1},\omega_{i+1}\ldots,\omega_m \in \{0,1\}} \int \left| g_{L,\beta,(\omega_1,\ldots,\omega_{i-1},0,\omega_{i+1}\ldots,\omega_m),h} - \right. \\
&\qquad\qquad\qquad\qquad\qquad\qquad\qquad\qquad\qquad \left. g_{L,\beta,(\omega_1,\ldots,\omega_{i-1},1,\omega_{i+1}\ldots,\omega_m),h} \right| \\
&\overset{(28)}{\leq} \frac{1}{2^m} \sum_{\omega_1,\ldots,\omega_{i-1},\omega_{i+1}\ldots,\omega_m \in \{0,1\}} 2Lh^{\beta+1} \int K \\
&= O\left(h^{\beta+1}\right).
\end{aligned}
$$

Here and in the sequel, the asymptotic comparators only hide constants (such as $\int K$ or $\int K^2$) and terms that depends on $L$ and $\beta$. All in all, by using Le Cam's lemma for product distribution and $\rho$-zCDP (Fact 5), and by leveraging Equation (13), with $\tau = \Omega(h^{2\beta+1})$,

$$
\inf_{\hat{\pi} \; \rho\text{-zCDP}} \sup_{\pi \in \Theta_{L,\beta}^{\mathrm{PSob}}} \mathbb{E}_{\mathbf{X} \sim \mathbb{P}_\pi^{\otimes n}, \hat{\pi}} \int_{[0,1]} (\hat{\pi}(\mathbf{X}) - \pi)^2 = \Omega\left(mh^{2\beta+1}\right)\left(1 - n\sqrt{\rho}O\left(h^{\beta+1}\right)\right). \tag{33}
$$

Setting $h \approx \left(n\sqrt{\rho}\right)^{\frac{-1}{\beta+1}}$ and $m = c/h$ for $c$ small enough concludes the proof by yielding a lower bound $\Omega\left(\left(n\sqrt{\rho}\right)^{-\frac{2\beta}{2\beta+1}}\right)$.

