# OpenReview forum: "About the Cost of Central Privacy in Density Estimation"
_TMLR — Accepted by TMLR_

### Review · Reviewer_uQLW · 2023-07-12

**Summary Of Contributions:**

The goal of the paper is to provide new lower and upper sample complexity bounds for non-parametric differentially private density estimation in the central (or "global") model. In terms of novelty, the bounds developed by the authors make the dependencies on $\epsilon$ explicit.

**Audience:**

Yes

**Broader Impact Concerns:**

No broader impact concerns---the paper is about differential privacy and the entire point is ethical concerns in CS/ML.

**Claims And Evidence:**

Yes

**Requested Changes:**

A bit more motivation of the problem and why having $\epsilon$ depend on $n$ is useful practically would be nice. Overall, however, this seems like a strong paper and I am happy with the current results.

**Strengths And Weaknesses:**

Overall, this seems like a solid paper. The technique developed by the authors have two benefits:
- they recover already known bounds without loss, hinting that the authors' techniques don't lose power due to their generality.
- the authors provide a lot of novel bounds, with lower and upper bounds often closely matching.
- the results are given both for pure DP and zCDP.
- the authors provide bounds under a general Lipschitz density assumption, and bounds under an assumption of periodic Sobolev densities that focuses on a less general class of smoother densities, but this allows the authors to obtain more refined bound.

Disclaimer: I have not had a chance to check the proofs carefully. That said, assuming the authors' results are right, I think this is a strong theory paper for DP density estimation. I am also not too familiar with previous work and unable to judge the novelty, and taking the authors' word on what has been and not been done.

The main and only complaint that I have is that the authors mention early on that "however, an important implicit hypothesis in this line of work is that ϵ, the parameter that decides how private the estimation needs to be, is supposed not to depend on the sample size. This hypothesis may seem disputable." I do not disagree with the authors, but can they motivate and discuss this point a bit more?

---

> ### Author Response · Authors · 2023-07-27
> **Answer to review**
>
> We thank the three reviewers for their analysis of our contributions and overall positive feedback on the article. We discuss below your specific remarks/questions/suggestions.
>
> - "... the authors mention early on that "however, an important implicit hypothesis in this line of work is that epsilon, the parameter that decides how private the estimation needs to be, is supposed not to depend on the sample size. This hypothesis may seem disputable." I do not disagree with the authors, but can they motivate and discuss this point a bit more?"
>
>
>  Primarily, we believe that working with fixed $\epsilon$ and deriving rates with respect to $n$ (as in a large part of the existing line of work) is a choice that restricts the considered problem and hides certain interaction phenomena between $n$ and $\epsilon$. Leaving $\epsilon$ as a free parameter, as we choose, allows for a better comprehension of the privacy/utility tradeoff and the different regimes. In particular, modern statistics under DP (like the work of Kamath et al. or Acharya et al. among others) often fit in this setup. It seems to us of particular interest to study when a substantial degradation of utility is due to privacy, that is when $\epsilon$ becomes smaller than a threshold that depends on $n$. This will be reworded, thank you for pointing out that the motivation for this choice can indeed be made clearer.

---

> > ### Comment · Action_Editors · 2023-07-27
> >
> > Can you point to an example to show the difference between your perspective and this other one? In particular, what hidden phenomena are uncovered? Using a specific problem to highlight this might be helpful.

---

> > > ### Author Response · Authors · 2023-08-04
> > >
> > > In the revision, we added a paragraph on page 2 that highlights the differences between fixed and varying epsilons. We motivated the regimes of non-increasing epsilons with the idea that one would like to leverage the increase in sample size ($n$) in order to increase each sample's privacy.

---

### Review · Reviewer_WJct · 2023-07-13

**Summary Of Contributions:**

The topic of this work is differentially-private (DP) non-parametric density estimation.
Specifically, the authors derive minimax rates for the density (w.r.t the Lebesgue measure on $[0,1]$) estimation problem (from $n$ iid samples) under:
- different notions of smoothness (Lipschitz, periodic Sobolev).
- different DP frameworks (pure/relaxed differential privacy with Laplace noise, concentrated differential privacy with Gaussian noise).


The analysis is conducted under a global privacy assumption where a central agent (aggregator) has access to the entire dataset and computes a DP estimator (as opposed to local privacy, where the aggregator only sees data which has already been DP blurred by each data holder separately).
Of particular interest, in the $\varepsilon$-DP setting, the authors investigate the influence of $\varepsilon$ on the minimax rates, highlighting the existence of a ``phase transition'' in the problem. When $\varepsilon$ becomes too small (compared to the sample size), DP becomes the bottleneck in the density estimation problem (high-privacy regime). On the other hand, when $\varepsilon$ is relatively large, DP does not impact the estimation rate.
Table~1 on p.3. compactly summarizes their results.

**Audience:**

Yes

**Claims And Evidence:**

Yes

**Requested Changes:**

- Please recall the definition of the Hamming distance when introduced on p.3.

- In Fact.1, the right hand side of the inequality reads as $\leq \frac{L^2}{\pi^{2 \beta}}$ where here $\pi$ if understood correctly is $3.1416...$. However the distribution in context is also $\pi$. Perhaps disambiguate $\pi$ in Fact 1 (adding a sentence for the reader).

- Add an explanation (in words) of global privacy before explaining local privacy.

- Typos:
  - p.2. remove space after trilemma in ``trilemma ,''
  - p.4 ``mechamisn''
  - p.7 ``Lebesque's measure''

**Strengths And Weaknesses:**

**Strengths**
1. The result gracefully recover known results in the literature ([Barber and Duchi, 2014] for Lipschitz densities).
2. In most cases, the authors obtain matching lower bounds, making their results particularly crisp.
3. The authors discuss optimality and sub-optimality of various classical lower bounding techniques (Fano,
Assouad, Tsybakov) for their problem.
4. The article is well-written and makes for a good read.

**Weaknesses**
1. The techniques employed to derive the rates appear to be standard.
2. In the flow of the paper, while global privacy is mentioned early in the paper, local privacy is currently explained before global privacy. This is a bit unnatural.

___

R. F. Barber and J. C. Duchi. Privacy and statistical risk: Formalisms and minimax bounds. arXiv preprint
arXiv:1412.4451, 2014

---

> ### Author Response · Authors · 2023-07-27
>
> We thank the three reviewers for their analysis of our contributions and overall positive feedback on the article. We discuss below your specific remarks/questions/suggestions.
>
> - "In the flow of the paper, while global privacy is mentioned early in the paper, local privacy is currently explained before global privacy. This is a bit unnatural." and "Add an explanation (in words) of global privacy before explaining local privacy."
>
> Thank you for pointing out this fact. Indeed, with this remark in mind, we now also feel that the reading flow is a bit unnatural. We will improve it by implementing the suggested modifications.
>
> -"In Fact.1, the right-hand side of the inequality reads as $\leq L^2 / \pi^{2 \beta}$ where here $\pi$ if understood correctly is $3.1416...$. However, the distribution in context is also $\pi$. Perhaps disambiguate in Fact 1 (adding a sentence for the reader)."
>
> We recognize a clumsy choice: the number $\pi$ should indeed not be confused with the density in the rest of the article. To avoid the confusion we will use "uppi" from the upgreek for $\pi$ here (the constant $\pi$ is not used elsewhere), and we will also add a small sentence to warn the reader.
>
> -"Please recall the definition of the Hamming distance when introduced on p.3."
>
> Thank you for detecting it, this will of course be fixed as well as the remaining typos.

---

### Review · Reviewer_P6Fj · 2023-07-24

**Summary Of Contributions:**

This paper studies the problem of density estimation under differential privacy. More specifically, the authors consider the Lipschitz and smooth distributions, and provide upper and lower bounds for the proposed differentially private density estimators. The main contribution of the paper is the new lower bound and upper bound results in different settings.

**Audience:**

Yes

**Claims And Evidence:**

Yes

**Requested Changes:**

Overall, the results are interesting and seems to be important. I have the following questions need to be addressed in the current paper:
1. Same results have been established in Barber & Duchi 2014 for Lipschitz distribution under $\epsilon$-DP. Then, why Theorem 1 is needed in the main paper since the $\rho$-zCDP results can be obtained by its relationship to $\epsilon$-DP.
2. Why do you consider the $\rho$-zCDP instead of the Renyi DP?
3. See point 2 in Weakness section.
4. In corollary 2, can you restate the results in terms of $\epsilon,\delta$ instead of $n^\gamma$. In this case, it is easy for the reader to compare the results to the results under $\epsilon$-DP.
5. The results in Table 1 about the Pure DP and Relaxed DP are also misleading. The relaxed DP result is missing the $\delta$ parameter which is the key difference when $\beta$ is large enough.


**Strengths And Weaknesses:**

Strengths of the paper:
1. It provides new lower bound results for the Lipschitz distribution under $\rho$-zCDP
2. It establishes new upper and lower bound results for smooth distributions
3. The upper bound results are (almost) optimal in several settings

Weaknesses of the paper:
1. The results in Theorem 1 seem to be some known results
2. It is unclear which part will be loss when using the general lower bound framework to establish the $\rho$-zCDP for the Lipschitz distribution
3. The statement in the corollary 2 with respect to the $\delta$ parameter under $(\epsilon,\delta)$-DP is unclear

---

> ### Author Response · Authors · 2023-07-27
>
> We thank the three reviewers for their analysis of our contributions and overall positive feedback on the article. We discuss below your specific remarks/questions/suggestions.
>
> -"Same results have been established in Barber \& Duchi 2014 for Lipschitz distribution under $\epsilon$-DP. Then, why Theorem 1 is needed in the main paper since the $\rho$-zCDP results can be obtained by its relationship to $\epsilon$-DP."
>
> The reason why this theorem is in the main paper is that it gives the upper bound for the \emph{pointwise difference} seminorm. In contrast, Barber \& Duchi 2014 frame it in terms on $L^2$ norm directly (which is in our paper a corollary of Theorem 1). Regarding the relations between $\rho$-zCDP and $\epsilon$-DP, it is true that changing the noise structure from Laplace to Gaussian was not necessary and that we could simply have used the algorithm class inclusion. This close relation between the result of Barber and Duchi and Thm 1 will be highlighted by rewording the few sentences following the theorem.
>
> -"Why do you consider the $\rho$-zCDP instead of the Renyi DP?"
>
> Looking at the results under Renyi DP can indeed be interesting. The reason why we looked at concentrated differential privacy was because of the use of the Gaussian mechanism in the upper bounds, and because of its close relation with concentrated differential privacy. With RDP however, we fear that the fact that the class of $(\alpha, \epsilon)$-RDP algorithms is not really stable under group privacy will make things harder for lower-bounds.
>
> -"It is unclear which part will be loss when using the general lower bound framework to establish the $\rho$-zCDP for the Lipschitz distribution"
>
> For the pointwise or the infinite norm risks, our proofs show that nothing is lost by using the general framework. For the $L^2$ risk on the other hand, using the general framework leads to a privacy overhead in the lower-bound of the order of $(n \sqrt{\rho})^{-4/3}$, which is suboptimal. This is to be found at page 28 of our article.
>
> -"In Corollary 2, can you restate the results in terms of $\epsilon, \delta$ instead of $n^\gamma$. In this case, it is easy for the reader to compare the results to the results under $\epsilon$-DP.} and {\em The results in Table 1 about the Pure DP and Relaxed DP are also misleading. The relaxed DP result is missing the $\delta$ parameter which is the key difference when $\beta$ is large enough."
>
> We will add a formulation in terms of $(\epsilon, \delta)$ in Corollary 2, and we will furthermore update Table 1 to reflect those changes.

---

### Comment · Action_Editors · 2023-07-27

Thanks to the authors for their responses to the reviewers. I would strongly recommend revising the paper, potentially with changes in blue, so that the reviewers can examine the revisions. OpenReview has a functionality for such revisions. There are 11 days before the reviewers are asked to submit a recommendation for the paper, so revising before then would be ideal.

---

> ### Author Response · Authors · 2023-08-04
>
> The revision has been submitted. As suggested, the changes compared to the last version are in blue. We tried to incorporate all the requested changes of the reviewers without significantly modifying the structure of the document. We also chose to use the term "central" differential privacy instead of "global".

---

> > ### Comment · Action_Editors · 2023-08-04
> >
> > Thanks authors. To the reviewers, please check out the changes made by the authors and ask any follow-up questions that might be applicable. Recall that you can enter your recommendation starting August 7.

---

### Author Response · Authors · 2023-08-30
**Camera ready and thank you**

We just submitted the camera ready version of the article. We also would like to thank the reviewers and the action editor for their valuable work.

---

### Decision · Action_Editors · 2023-08-23

**Recommendation:** Accept as is

**Comment:**

The authors provide new lower bounds under central differential privacy for various estimation tasks. Reviewers agreed these were interesting. There were fairly few questions or concerns, but the authors dispatched them with alacrity. Assuming soundness, which the reviewers vouch for, then the paper should be accepted.

**Audience:**

The (theory of) differential privacy community is present in the TMLR audience, and the reviewers agree that the results are new and interesting additions to the literature.

**Claims And Evidence:**

The paper gives mathematical proofs of all claims, and to the best of the the reviewers' estimates, they appear to be sound.